# FakeMark: Deepfake Speech Attribution With Injected System Signatures

## Abstract

Deepfake speech attribution remains challenging for existing solutions. Classifier-based solutions often fail to generalize to domain-shifted samples, and watermarking-based solutions can be compromised by distortions like codec compression or malicious removal attacks. To address these issues, we propose FakeMark, a novel watermarking framework that injects artifact-correlated watermarks associated with deepfake generation systems rather than pre-assigned bitstring messages. These watermarks, referred to as system signatures, enable a classifier-based decoder to attribute the source system by jointly leveraging the injected signature and intrinsic deepfake artifacts, remaining effective even if one of these cues is elusive or removed. Experimental results show that FakeMark improves generalization to cross-dataset samples where classifier-based solutions struggle and maintains high accuracy under various distortions where conventional watermarking-based solutions fail. Speech samples are available at `https://fakemark-demo.github.io/fakemark-demo/`.[1]

## 1 Introduction

Attributing deepfake speech (illustrated in Figure 1 (a)) requires identifying the source system that generated the synthetic sample after it has been flagged as deepfake (Müller et al., 2022; Klein et al., 2025). This capability is essential for applications such as generative system identification and data provenance analysis (Xu et al., 2025; Fernandez et al., 2023), enabling auditing, accountability, and responsible deployment of speech generation technologies. Most attribution methods train deep-neural-network-based classifiers (illustrated in Figure 1 (b)) for system identification (Sun et al., 2023; Wang et al., 2025). They often require to be trained in a discriminative manner with data generated by a rich variety of speech synthesis systems to capture artifact differences. However, such classifiers are known to be sensitive to domain shift and often fail on deepfakes from unseen systems (Bhagtani et al., 2024; Chen et al., 2025b), as their performance is fundamentally constrained by the variability present in the training data.

Recently, watermarking-based methods become popular as an alternative solution to the attribution task (Cho et al., 2022; Li et al., 2025b; Yang et al., 2025). These approaches train a pair of watermark generator and decoder (illustrated in Figure 1 (c)), where the generator injects a watermark message into the carrier speech that is later extracted by the decoder; attribution is achieved by mapping the extracted message to its pre-assigned system label. Although watermarking models have demonstrated high accuracy on various benchmarks (Liu et al., 2024b; Roman et al., 2024), they are not perfectly robust to distortions and removal attacks (Yang et al., 2024; Kassis & Hengartner, 2025). In its application to speech, for example, generators are trained to inject watermarks that are inaudible to the human ear. Yet decoders often struggle under neural codec transmissions (Juvela & Wang, 2025; O'Reilly et al., 2025), whose training objective is compression and high-fidelity reconstruction of audio signals (Miller et al., 1999). In deepfake related tasks such as detection (Wu et al., 2025), classifier-based solutions remain robust under neural codecs since deepfake artifacts are preserved to some extent, whereas watermark decoders suffer notable degradation as the injected messages are removed during compression.

Presented in this paper is our attempt to address the above challenges for robust deepfake speech attribution. Specifically, we ask the following research question: **Can we enhance deepfake trace-**

---

[1]Codes and model checkpoints will be released upon acceptance.

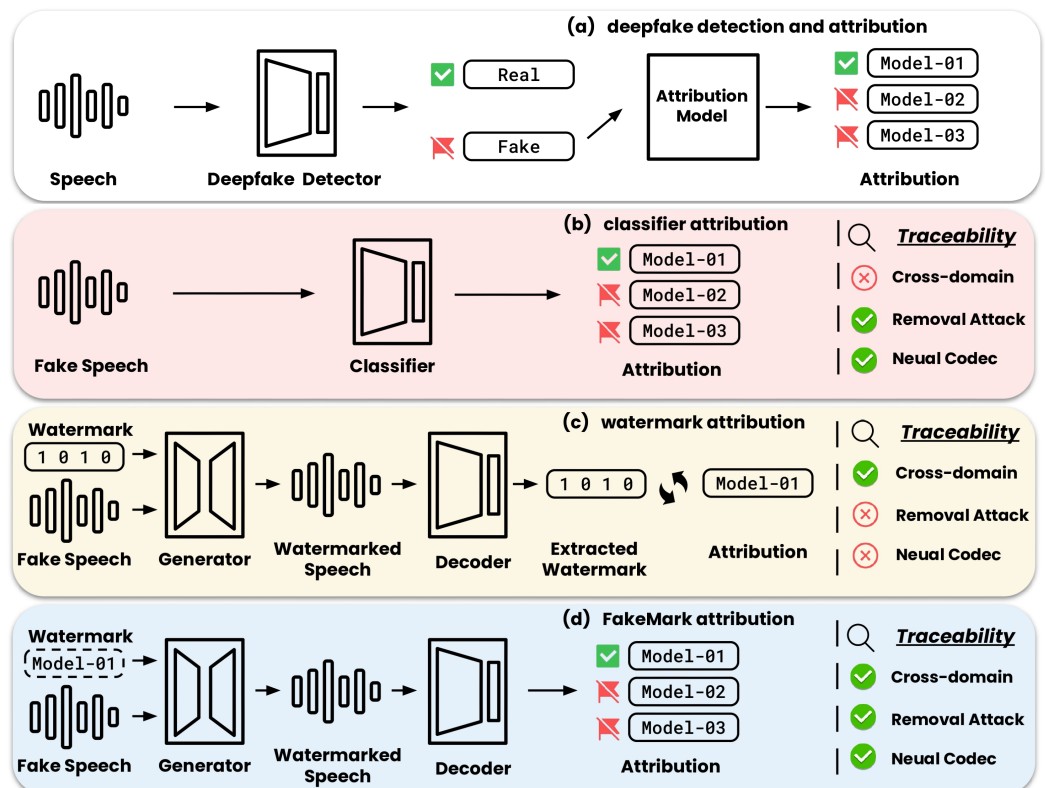

Figure 1: Illustrations of (a) usage of deepfake detection and attribution, (b) classifier-based attribution, (c) watermarking-based attribution, and (d) our proposed FakeMark.

**ability by injecting artifact-correlated watermarks?** We hypothesize that correlating watermarks with deepfake artifacts can provide improved 1) **robustness towards distortions** by enabling the decoder to perform attribution through acoustic artifacts when the injected signature is removed; and 2) **generalization performance** by introducing a watermark generator to assist the typically standalone classifier when seen artifact patterns are absent.

To answer the question, we propose FakeMark (illustrated in Figure 1 (d)). The FakeMark decoder operates in standard classifier-based attribution settings, while its generator allows content owners to proactively enhance deepfake traceability prior to distribution. Unlike bitstring messages used in traditional watermarking approaches, FakeMark generator injects *system signatures*, which are watermarks correlated with system-specific artifacts. These signatures are designed to survive distortions and complement intrinsic deepfake artifacts for robust attribution. Our main contributions are summarized as follows:

- We introduce FakeMark, a novel watermarking framework for deepfake speech attribution. The FakeMark generator injects artifact-correlated system signatures as watermarks, allowing its decoder to map either the artifacts or the watermarks to their source system.
- We present the first systematic evaluation of deepfake speech attribution using both watermarking-based and classifier-based models. We evaluate FakeMark against these baselines on common datasets and under diverse distortions, showing that it improves attribution robustness and generalization in challenging scenarios.

## 2 RELATED WORKS

**Speech generation** typically follows two paths: text-to-speech (TTS) and voice conversion (VC). In modern TTS, an acoustic model maps the input text (or its derived linguistic features) to an intermediate acoustic representation that is either continuous valued hidden feature vectors or discrete

tokens. A neural vocoder is then used to synthesize the speech waveform (Tan et al., 2021). VC follows a partially similar design: it takes an input waveform from a source speaker and renders the same content in a target speaker's voice. The term *artifacts* denotes deviations of synthesized speech from natural speech. Common audible artifacts include (i) alignment errors between text and predicted acoustics that cause word skipping or repetition (Zen et al., 2009); (ii) insufficient modeling of prosody (e.g., incorrect pitch accent (Łańcucki, 2021)), expressiveness (e.g., flat intonation (Liu et al., 2021; Mahapatra et al., 2025)) , and speaker characteristics (e.g., a voice perceptually dissimilar to the target speaker (Chen et al., 2025a; Pan et al., 2022)); and (iii) vocoder artifacts such as buzziness or high-frequency noise (Bak et al., 2023; Sun et al., 2023).

**Deepfake attribution.** Depending on the specific architectures used, it has been reported that different acoustic models (Bhagtani et al., 2024; Chen et al., 2025c) and vocoders (Sun et al., 2023; Deng et al., 2024) leave distinctive artifacts that can be leveraged for deepfake attribution. Solutions to the task naturally involve collecting samples from various TTS and VC systems (Müller et al., 2024; Chen et al., 2025b). These samples are then used to train multiclass classifiers to predict the acoustic models or vocoders used for their generation (Klein et al., 2024). However, such supervised training scheme can sometimes cause classifiers to exploit undesired differences in the training data, leading to poor generalization performance on unseen samples. This includes samples generated by seen systems but trained with different languages (Marek et al., 2025) or speakers (Klein et al., 2025), and samples with subtle artifacts like unnatural silences (Chen et al., 2025c) or even generated by the same system with different weights (Stan et al., 2025). Alternative strategies to achieve generalization include estimating model confidence on unseen samples (Klein et al., 2025) or measuring sample similarities in latent space, akin to verification tasks (Negroni et al., 2025).

**Speech watermarking** models are designed to inject and extract bitstring messages within a speech signal (Li et al., 2025b; Liu et al., 2024b). Depending on the information carried in the message, these models are versatile for various tasks (Miller et al., 1999). For example, a watermark can encode a compressed version of the original signal for self-recovery (Quiñonez-Carbajal et al., 2024), or it can be assigned to different users for attribution and copyright protection (Roman et al., 2024; Liu et al., 2024a). In deepfake-related applications, different bitstrings can be assigned to real and fake samples for detection (Wu et al., 2025; Roman et al., 2024) or to counter malicious deepfake manipulations (Li et al., 2025a; He et al., 2025). Beyond bitstring messages, the presence of a watermark itself can represent a zero-bit message indicating whether a sample is real or fake (Juvela & Wang, 2024; Roman et al., 2025). Previous studies have reported that watermarking-based models can be vulnerable to distortions such as neural codecs (Défossez et al., 2023; Ju et al., 2024), malicious forgery, or removal attacks (Yang et al., 2024; Liu et al., 2024b). Common strategies to enhance robustness include using codec-based data augmentation during training (Juvela & Wang, 2025) and injecting watermarks into deep latent representations of the speech signal (Ji et al., 2025b).

## 3 FAKEMARK

We describe FakeMark pipelines for system signature (i.e., watermark) injection and extraction in Sec. 3.1. Objectives used to train the system modules are detailed in Sec. 3.2.

### 3.1 PIPELINE

As illustrated in Figure 2, FakeMark takes two inputs during watermark injection: the speech signal $s \in \mathbb{R}^T$ and the watermark index $w \in \{1, \ldots, C\}$, where each index corresponds to a learned watermark embedding stored in the message processor, and $T$ is the number of waveform sampling points and $C$ is the total number of deepfake systems. It outputs a watermarked signal $s_w \in \mathbb{R}^T$ that carry the watermark and has the same dimensionality as the input signal. During watermark extraction, FakeMark takes $s_w$ and predicts a watermark index $w'$.

The injection process involves four stages:

1. Given the input watermark index $w$, the message processor returns the corresponding embedding vector $e_w \in \mathbb{R}^H$ to the generator, where $H$ is the latent dimension.

2. Given the input signal $s$, the generator down-samples it and extracts a compact latent representation $H_s \in \mathbb{R}^{\lfloor \frac{T}{\alpha} \rfloor \times H}$, where $\alpha$ is the downsampling factor.

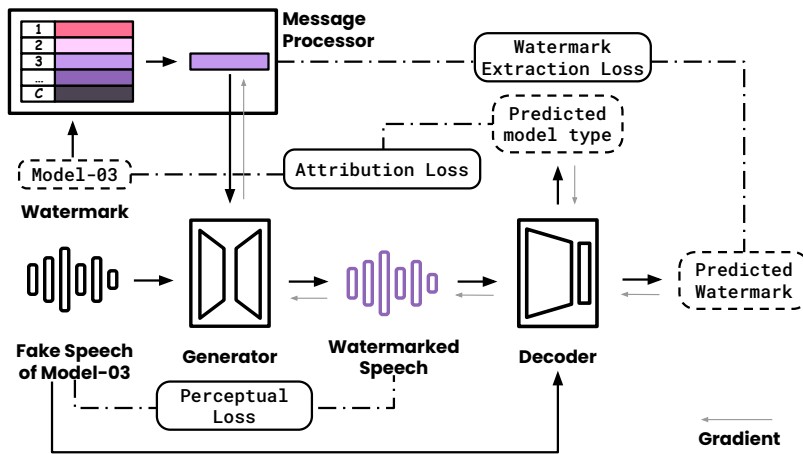

Figure 2: Training pipeline of FakeMark.

3. Given the input watermark embedding $\boldsymbol{e}_w$, the generator repeats it along the time axis to form $\boldsymbol{E}_w \in \mathbb{R}^{\lfloor \frac{T}{\alpha} \rfloor \times H}$, then applies voice-activity detection (VAD) to obtain a binary mask $\boldsymbol{m} \in \{0,1\}^{\lfloor \frac{T}{\alpha} \rfloor}$ indicating speech-active frames, and computes the watermark latent $\boldsymbol{H}_w = \boldsymbol{m} \odot \boldsymbol{E}_w$, where $\boldsymbol{H}_w \in \mathbb{R}^{\lfloor \frac{T}{\alpha} \rfloor \times H}$.

4. The generator up-samples $\boldsymbol{H}_s + \boldsymbol{H}_w$ and outputs the watermark signal $\boldsymbol{\delta}_w \in \mathbb{R}^T$. The final watermarked signal is obtained as $\boldsymbol{s}_w = \boldsymbol{s} + \boldsymbol{\delta}_w$, where $\boldsymbol{s}_w \in \mathbb{R}^T$.

Following the injection pipeline, we explore two generator architectures:

- FakeMark$^A$: follows an encoder-decoder architecture that processes speech waveforms, as used in AudioSeal (Roman et al., 2024);
- FakeMark$^T$: follows an encoder architecture that processes spectrogram features, as used in Timbre (Liu et al., 2024a). In this setting, $\boldsymbol{H}_s$ is the linear-scale spectrogram obtained via Short-Time Fourier Transform (STFT). Final waveforms are obtained via inverse STFT.

The extraction process involves two stages:

1. During training, given the input watermarked signal $\boldsymbol{s}_w$, the decoder applies a series of transformations to obtain a distorted version input $\boldsymbol{s}'_w$. This strategy ensures robustness of the watermark injection and extraction. Full list of the used transformations and their settings are provided in Appendix A.2. These transformations are disabled during inference.

2. The decoder extracts sequence-level feature from the input waveform and predicts the class probabilities $\mathbf{p} \in [0,1]^C$ over $C$ watermark types; the extracted watermark is obtained as $w' = \arg\max_{i \in \{1,...,C\}} p_i$.

We use a common decoder architecture that consists of a pre-trained SSL front-end (Pratap et al., 2024) and a fully connected back-end classifier. Detailed architectures are provided in the Appendix A.3.

## 3.2 TRAINING OBJECTIVES

All FakeMark modules are optimized with three classes of objectives: 1) attribution loss, to ensure the ability of distinguishing different types of artifacts; 2) extraction loss, to maximize the successful injection and extraction of watermarks; and 3) perceptual loss, to minimize the perceptual distortion between original and watermarked signals. They are detailed below.

**Attribution loss** differentiates FakeMark from conventional watermarking approaches, where decoder is trained solely to extract bitstring messages from arbitrary inputs. During FakeMark training,

we compute the cross-entropy between the ground-truth deepfake system label and the decoder's predicted class probabilities over an *unwatermarked* clean signal as attribution loss. This objective is similar to training classifier-based attribution models (Klein et al., 2024; Sun et al., 2023), where the goal is to capture distinct characteristics of deepfakes generated by different systems. The back-propagated attribution loss encourages the decoder to distinguish various types of deepfake artifacts and this learned behavior further guides the watermark embeddings from the message processor to correlate with these artifacts. As a result, each watermark embedding encodes the artifact patterns learned from all samples of a specific deepfake system in the training set.

**Watermark extraction loss** ensures that the generator injects watermarks that can be reliably recognized by the decoder. Similar to the training of conventional watermarking models, a *random* watermark index is sampled and processed to retrieve a watermark embedding, which is then used for watermarked injection. The decoder predicts class probabilities from this *watermarked* signal, and the watermark extraction loss is computed as the cross-entropy between the ground-truth watermark index and the predicted distribution.

By jointly training FakeMark with both attribution and watermark extraction losses, the message processor learns to align watermark embeddings with the deepfake artifacts they represent (illustrated in Figure 4), and the generator-decoder pair learns to robustly inject and detect these watermarks. During inference, with the watermark index always chosen to match the ground-truth deepfake system, FakeMark can attribute the source system using both acoustic artifacts and the watermarks. This ensures effective attribution even if one of these cues is compromised.

**Perceptual losses** promote the imperceptibility of watermarks and the naturalness of watermarked signals. This is achieved by refining the watermarked signal with HiFi-GAN–style losses (Kong et al., 2020), which include a Mel-spectrogram reconstruction loss to enforce the spectral similarity and adversarial discriminator losses to improve speech fidelity.

Additionally, we follow Roman et al. (2024) to refine the watermark signals generated by both generator architectures. We apply $l_1$ loss and loudness on the watermark signal to decrease its intensity and ensure its robustness towards distortions. We further use a frequency magnitude loss to align the averaged spectral envelope of the watermark signal with that of the clean signal, promoting perceptual similarity and ensuring the watermarks remain less audible.

## 4 EXPERIMENTS AND RESULTS

We perform both in-domain evaluation and cross-dataset evaluation. FakeMark is compared against recent baselines on both clean and distorted signals. In addition to attribution accuracy, we also assess the speech quality of watermarked signals.

### 4.1 EXPERIMENTAL SETUP

**Datasets.** We use the MLAAD_v5 dataset (Müller et al., 2024) for training and in-domain evaluation. Following the source tracing challenge protocol (Müller, 2024), our training set comprises 24 TTS systems covering eight languages. To mitigate the influence of undesirable artifacts related to language or speaker (Klein et al., 2024), we group systems with identical architectures into a single class. The resulting training set contains 12 classes, 9 of which appear in the evaluation set. For cross-dataset evaluation, we collected samples generated by five of these systems from ASVspoof5 (Wang et al., 2024) and TIMIT-TTS (Salvi et al., 2023) datasets. Both evaluation sets were randomly sampled with an equal number of files per class. All evaluated system architectures are included in the training data; evaluations on other architectures are beyond the scope of this work. Dataset details are provided in Appendix A.4.

**Baseline systems.** We compare our FakeMark with recent and remarkable watermarking models: AudioSeal (Roman et al., 2024) and Timbre (Liu et al., 2024a), and a ResNet34-based classifier (Klein et al., 2025) that takes STFT spectrograms as input. Additionally, we train an SSL-based classifier with the same architecture as FakeMark decoder (denoted MMS-300M) to isolate the impact of the watermarking scheme. All baselines were trained with the same training set and augmentation strategy as FakeMark. Watermark message length for AudioSeal and Timbre is set to

4 (equivalent to $2^4$ unique bitstrings)-the minimum capacity required to cover 12 classes. Full model configurations and implementation details are in Appendix A.5.

**Evaluation metrics.** Objective evaluation for both attribution and audio quality are performed to evaluate FakeMark and baselines:

- For attribution performance, we report accuracy result. Predicted class for AudioSeal and Timbre is the class whose assigned 4-bit message has the shortest Hamming distance to the decoder output (Roman et al., 2024; Liu et al., 2025). For FakeMark and classifier-based models, predicted class is the class with the highest decoder probability.

- For audio quality assignment of watermarked signals, we use four different metrics: Scale Invariant Signal to Noise Ratio (SI-SNR) for evaluating noise-level of watermarks; PESQ to evaluate speech quality for telecom-like scenarios (Rix et al., 2001); ViSQOL for assessing perceptual quality for network-based scenarios (Hines et al., 2012); Production Quality (PQ) to estimate the clarity and fidelity of watermarked signals (Tjandra et al., 2025). Unlike the previous three metrics, PQ does not require clean reference signals.

**Distortions and attacks.** To evaluate system robustness against distortions and watermark removal attacks, we apply a set of transformations previously shown to have a noticeable impact on either watermark extraction (O'Reilly et al., 2025; Yao et al., 2025; Yang et al., 2024) or deepfake detection (Wu et al., 2025) in the literature. They are applied to the watermarked signals for FakeMark, AudioSeal, and Timbre, and directly to the input signals for ResNet34 and MMS-300M, include:

- Signal processing-based transforms: Pitch shift, playback speed change, and additive noise from MUSAN (Snyder et al., 2015);

- Neural Codec-based waveform compression and regeneration: SpeechTokenizer (Zhang et al., 2024a), FACodec (Ju et al., 2024; Zhang et al., 2024b), and WavTokenizer (Ji et al., 2025a);

- Neural Vocoder-based waveform regeneration: HiFi-GAN (Kong et al., 2020), Vocos (Siuzdak, 2024), and BigVGAN (Lee et al., 2023).

- Watermark removal attacks: Overwriting (Yao et al., 2025), where publicly available watermarking models (WavMark (Chen et al., 2023) and SilentCipher (Singh et al., 2024)) are applied sequentially to overwrite the existing watermark; and Averaging (Yang et al., 2024), where an average watermark is estimated from the evaluated models and then subtracted from the watermarked signals. We did not apply Averaging attack with Timbre model because its generator directly outputs the watermarked signal rather than estimating a separate watermark.

Details of distortions and attacks can be found in Appendix A.6.

## 4.2 RESULTS

We report deepfake attribution performance in this section, including in-domain evaluation results in Sec. 4.2.1 and cross-dataset evaluation results in Sec. 4.2.2. Speech quality evaluation and additional analysis are reported in Sec. 4.2.3 and Sec. 4.2.4.

### 4.2.1 IN-DOMAIN EVALUATION

Table 1 presents attribution accuracy for FakeMark and baselines on the MLAAD_v5 test set. Rows represent accuracies under different distortions. Cells are color-coded in grayscale by row: darker shades indicating lower accuracy and lighter shades indicating higher accuracy. We summarize observations related to our research question below.

**FakeMark is robust to strong watermark removal distortions.** When no distortion is applied, all models achieve near-perfect accuracy (above 0.97). Across most distortions—except background noise and WavTokenizer—both FakeMark variants maintain high attribution accuracy (above 0.80). In contrast, AudioSeal accuracy drops dramatically under codec (0.09–0.17) and vocoder (0.09–0.28) reconstructions, which is expected given that these distortions are known strong watermark removers (O'Reilly et al., 2025; Juvela & Wang, 2025). Although FakeMark[4] shares the same

Table 1: Attribution accuracy results on MLAAD_v5 test set across distortions and attacks.

| System / Distortion | | Proposed Method | | Watermarking Baselines | | Classifier Baselines | |
|---|---|---|---|---|---|---|---|
| | | FakeMark$^A$ | FakeMark$^T$ | AudioSeal | Timbre | MMS-300M | ResNet34 |
| | None | 1.00 | 1.00 | 1.00 | 1.00 | 1.00 | 0.97 |
| Signal Processing | Pitch | 0.82 | 0.96 | 0.80 | 0.96 | 0.27 | 0.88 |
| | Speed | 0.99 | 1.00 | 0.85 | 0.97 | 1.00 | 0.92 |
| | Noise | 0.63 | 0.74 | 0.72 | 0.60 | 0.80 | 0.50 |
| Codec | SpeechTokenizer | 0.85 | 0.99 | 0.10 | 0.94 | 0.92 | 0.88 |
| | FACodec | 0.91 | 0.85 | 0.17 | 0.82 | 0.92 | 0.79 |
| | WavTokenizer | 0.33 | 0.47 | 0.09 | 0.19 | 0.39 | 0.71 |
| Vocoder | HiFi-GAN | 0.91 | 1.00 | 0.09 | 1.00 | 0.94 | 0.92 |
| | Vocos | 0.98 | 1.00 | 0.12 | 1.00 | 0.98 | 0.97 |
| | BigVGAN | 0.99 | 1.00 | 0.28 | 1.00 | 1.00 | 0.97 |
| Removal Attack | Overwriting | 0.97 | 0.91 | 0.98 | 1.00 | 0.97 | 0.69 |
| | Averaging - AudioSeal | 1.00 | 1.00 | 0.82 | 1.00 | 1.00 | 0.97 |
| | Averaging - FakeMark$^A$ | 1.00 | 1.00 | 0.99 | 1.00 | 1.00 | 0.97 |
| | Averaging - FakeMark$^T$ | 1.00 | 1.00 | 1.00 | 1.00 | 1.00 | 0.97 |

generator architecture as AudioSeal, its decoder can still leverage deepfake artifacts for attribution, yielding performance that is similar to that of the MMS-300M classifier.

Our baseline Timbre model demonstrates unexpectedly robust performance across distortions previously reported as vulnerabilities (O'Reilly et al., 2025; Özer et al., 2025). This is likely due to retraining with additional augmentations, including a codec method named EnCodec (Défossez et al., 2023).

**FakeMark demonstrates strong robustness to averaging-based removal attacks.** It maintains perfect attribution accuracy (1.0) whether the averaged watermark is derived from the attacked model itself or estimated using a separate model. In contrast, overwriting attacks cause noticeable degradation (from 1.0 to 0.97 for FakeMark$^A$ and to 0.91 for FakeMark$^T$). We hypothesize that this occurs because overwriting disrupts artifact patterns, as suggested by the performance drops in classifier-based baselines (from 1.0 to 0.97 for MMS-300M and from 0.97 to 0.69 for ResNet34).

**Additional discussion: models process spectrogram features are generally more robust.** Table 1 shows that attribution is easily solved under clean conditions. Even with distortions, most models—except AudioSeal—maintain reliable performance in many scenarios. We also notice that models process spectrogram features are more robust to distortions compared to their counterparts. Watermarking models such as FakeMark$^T$ and Timbre achieve perfect accuracies (1.0) under neural vocoders. The ResNet34 is the only solution that does not reach perfect performance under clean conditions (0.97); however, its lowest accuracy (0.50 under Noise) remains noticeably higher than the MMS-300M's lowest results (0.27 under Pitch shift and 0.39 under WavTokenizer).

Almost all tested models appear sensitive to signal processing–based distortions but relatively more robust to other types of distortions. This is expected, as signal processing transforms directly modify the speech signal and thus alter artifact patterns. By contrast, reconstruction-based distortions primarily regenerate the waveform together with artifacts and, in some cases, watermarks. In the next section, we show that attribution becomes more difficult when the artifact patterns are not present in the training set (i.e., unseen), particularly for the two classifier-based baselines.

### 4.2.2 CROSS-DATASET EVALUATION

Table 2 present cross-dataset evaluation of attribution accuracy for FakeMark and baseline models. Results are presented in a similar format as Table 1. We summarize observations related to our research question below.

**FakeMark performs robustly under domain shift.** Under clean conditions, both FakeMark variants and the watermarking baselines achieve perfect accuracy (1.0), this is consistent with their in-domain results in Table 1. The two classifier-based models perform poorly (0.07 for MMS and 0.12 for ResNet34), likely due to their limited generalization to unseen artifact patterns—even those produced by TTS architectures seen during training. The two classifiers give similar performance

Table 2: Attribution accuracy results on ASVspoof5 and TIMIT-TTS datasets across distortions and attacks.

| Distortion (System) | Proposed Method | | Watermarking Baselines | | Classifier Baselines | |
|---|---|---|---|---|---|---|
| | FakeMark$^A$ | FakeMark$^T$ | AudioSeal | Timbre | MMS-300M | ResNet34 |
| None | 1.00 | 1.00 | 1.00 | 1.00 | 0.07 | 0.12 |
| **Signal Processing** | | | | | | |
| Pitch | 0.80 | 0.97 | 0.72 | 0.96 | 0.00 | 0.10 |
| Speed | 0.99 | 0.99 | 0.78 | 0.98 | 0.06 | 0.11 |
| Noise | 0.58 | 0.61 | 0.65 | 0.62 | 0.03 | 0.05 |
| **Codec** | | | | | | |
| SpeechTokenizer | 0.58 | 0.94 | 0.07 | 0.90 | 0.07 | 0.10 |
| FACodec | 0.87 | 0.32 | 0.08 | 0.85 | 0.08 | 0.05 |
| WavTokenizer | 0.06 | 0.02 | 0.03 | 0.21 | 0.07 | 0.07 |
| **Vocoder** | | | | | | |
| HiFi-GAN | 0.88 | 1.00 | 0.08 | 1.00 | 0.07 | 0.11 |
| Vocos | 0.98 | 1.00 | 0.09 | 1.00 | 0.03 | 0.11 |
| BigVGAN | 1.00 | 1.00 | 0.19 | 1.00 | 0.06 | 0.11 |
| **Removal Attack** | | | | | | |
| Overwriting | 0.95 | 0.70 | 0.98 | 0.92 | 0.05 | 0.01 |
| Averaging - AudioSeal | 1.00 | 1.00 | 0.93 | 1.00 | 0.06 | 0.12 |
| Averaging - FakeMark$^A$ | 1.00 | 1.00 | 1.00 | 1.00 | 0.06 | 0.11 |
| Averaging - FakeMark$^T$ | 1.00 | 1.00 | 1.00 | 1.00 | 0.06 | 0.11 |

under distortions and attacks, not because their robustness improves in these conditions, but rather because their clean-condition accuracy is already very low.

As shown in Table 2, the performance of FakeMark and watermarking baselines degrades when input signals are distorted, but the trends remain similar to those in Table 1, with a slight drop in overall accuracy. Given that MMS-300M fails on this domain-shifted data (highest accuracy 0.08), the robustness of FakeMark decoder (above 0.80 under most distortions) can be attributed primarily to the watermarks injected by its generator. Unlike classifier-based models, FakeMark decoder is influenced more by distortions applied to the carrier signal than by the carrier itself.

**Watermarks injected by FakeMark are robust to averaging-based attacks.** Though the last three rows of Table 1 might suggest that this robustness comes from artifact seen during training, the cross-dataset results in Table 2 show that both FakeMarks remain equally robust even when such artifacts are absent. This indicates that the robustness primarily stems from the injected watermark itself, which is correlated with acoustic artifacts rather than fixed waveform patterns that averaging attacks mainly target (Yang et al., 2024).

**Additional discussion on watermarking in deepfake attribution.** Both FakeMark variants and Timbre outperform the two classifiers in nearly all test cases across in-domain (Table 1) and cross-dataset (Table 2) evaluations. Beyond the robustness provided by the system design and training strategies, it is important to note that these solutions are designed for different application scenarios. Classifier-based solutions are passive and require no prior knowledge of the input signal, whereas watermarking-based solutions are proactive and require a watermark to be injected into the decoder input in advance. In the following sections, we further assess the impact of the injected watermarks on speech quality (Sec. 4.2.3) and decoder performance (Sec. 4.2.4).

### 4.2.3 EVALUATION ON SPEECH QUALITY AND INTELLIGIBILITY

We evaluate the quality and intelligibility of watermarked signals. Results are presented in Table 3. Our observations are summarized below.

Table 3: Comparison of speech quality and intelligibility on watermarked speech signals generated by FakeMark and watermarking-based baselines.

| | System | SI-SNR ↑ | PESQ ↑ | ViSQOL ↑ | PQ ↑ |
|---|---|---|---|---|---|
| **Baselines** | AudioSeal | 36.49 | 4.55 | 4.98 | 6.78 |
| | Timbre | 21.79 | 2.97 | 4.20 | 5.67 |
| **Proposed** | FakeMark$^A$ | 35.34 | 3.79 | 4.81 | 6.62 |
| | FakeMark$^T$ | 19.14 | 3.09 | 4.64 | 6.50 |

**FakeMark$^A$ achieves second in speech quality.** The FakeMark$^A$ performs second only to AudioSeal. Its relatively high SI-SNR (35.34 dB) suggests that the injected watermark has low energy compared to the clean carrier. For other speech quality and fidelity metrics, AudioSeal is the only system achieving a PESQ score above 4 (4.55), while FakeMark$^A$ is slightly lower in ViSQOL (4.98 vs. 4.81) and PQ (6.78 vs. 6.62).

**Trade-off between robustness and speech quality.** We observe that watermarks injected through spectrogram features (Timbre and FakeMark$^T$) introduce more audible distortion than waveform-based methods (AudioSeal and FakeMark$^A$). Their worse speech quality contrasts with observations on attribution performance in Sec. 4.2.1, and suggests a trade-off between attribution robustness and speech quality. The consistent near-perfect performance of Timbre and FakeMark$^T$ is achieved through stronger, more perceptually noticeable watermarks that can survive multiple distortions (shown in Appendix A.9). In contrast, AudioSeal's less perceptible watermark introduces minimal distortion to the carrier but is the most vulnerable among the evaluated models. Our proposed FakeMark$^A$ provides strong watermark injection while maintaining relatively high speech quality.

### 4.2.4 IMPACT OF WATERMARKS ON ATTRIBUTION

In this section, we examine the extent to which the injected watermarks improve deepfake traceability. We compare FakeMark detector's performance on non-watermarked, clean signals with the results from Tables 1 and 2, where decoder inputs are watermarked signals. Results are presented in Table 4. Our observations are summarized below.

Table 4: Attribution accuracy results of FakeMark detector under different watermarking conditions.

| Generator | Test set / Distortion — Condition | In-domain | | Cross-dataset | |
|---|---|---|---|---|---|
| | | No watermark | Table 1 | No watermark | Table 2 |
| FakeMark$^A$ | None | 1.00 | 1.00 | 0.06 | 1.00 |
| | Others averaged | 0.81 | 0.88 | 0.04 | 0.82 |
| FakeMark$^T$ | None | 0.99 | 1.00 | 0.06 | 1.00 |
| | Others averaged | 0.85 | 0.92 | 0.04 | 0.81 |

**The injected watermarks improve deepfake traceability.** Similar to the classifier baselines in clean conditions, the standalone FakeMark decoder achieves near-perfect accuracy (above 0.99) on the in-domain test set, but drops to 0.81 (FakeMark$^A$) and 0.85 (FakeMark$^T$) under distortions. Adding watermarks improves attribution accuracy for both variants (0.88 for FakeMark$^A$ and 0.92 for FakeMark$^T$). The gains are particularly notable in cross-dataset evaluations, where the standalone decoder performs poorly (below 0.1) but exceeds 0.8 once watermarks are injected.

The right side of in Table 4 show that, despite sharing similar generation architectures with the training systems, differences in data, language, and speakers make cross-dataset artifacts largely unrecognizable to the decoder. While the shared architecture still allows FakeMark to function without retraining or introducing new class labels, in the next section, we consider a more challenging open-set setting where deepfakes signals are generated by entirely different architectures.

### 4.3 APPLICATION TO UNSEEN ARCHITECTURES

In this section, we show how FakeMark can be applied to signals generated by architectures not included during training (i.e., unseen). The setup for adapting FakeMark to these unseen systems without retraining is described in Sec. 4.3.1. Results are reported in Sec. 4.3.2.

### 4.3.1 SETUP

**Use case.** We assume content owners have access to the target signals and are aware of their generation systems before distribution. These signals are then watermarked using the system signature of the training system whose centroid is closest to the unseen target system in embedding space (detailed and visualized in Appendix A.7). The watermarked signals are produced by the FakeMark generator and distributed in the same manner as in our previous experiments. Upon receiving the possibly distorted signals, content owners use the FakeMark decoder for further processing.

**Verification and metric.** As described in Sec. 4.1, FakeMark decoder is trained to output probabilities over 12 classes. Signals generated by systems outside these 12 classes cannot be evaluated using a standard accuracy metric. Following Negroni et al. (2025), we instead compute the cosine similarity between the decoder's output embedding for a reference signal and that for a test signal. High similarity scores indicate that the two signals share the same origin (i.e., both were injected with the same system signature), whereas low scores suggest otherwise. During verification, the watermarked test signal may or may not originate from the same system architecture as the reference signal. The obtained similarity scores are then used to calculate an equal error rate (EER), a widely adopted metric in verification tasks (Negroni et al., 2025; Chung et al., 2018).

**Datasets.** We collected speech samples from three systems in ASVspoof5 and TIMIT-TTS: ToucanTTS (Lux et al., 2022), GradTTS (Popov et al., 2021), and gTTS (Durette, 2022). None of these systems are included in the MLAAD_v5 training set. Dataset details are reported in Appendix A.7.

### 4.3.2 RESULTS

Table 5 presents verification results for FakeMark on the unseen ASVspoof5 + TIMIT-TTS test set. The EER values reflect FakeMark decoder's discriminability under each distortion, with lower EER indicating better verification performance. Our observations are summarized below.

Table 5: Equal error rate (EER, %) results for FakeMark across distortions and attacks.

| | Distortion | FakeMark$^A$ | FakeMark$^T$ |
|---|---|---|---|
| Signal Processing | Pitch | 8.33 | 5.33 |
| | Speed | 2.33 | 0.00 |
| | Noise | 34.67 | 24.00 |
| Codec | SpeechTokenizer | 24.67 | 3.33 |
| | FACodec | 8.00 | 26.00 |
| | WavTokenizer | 45.67 | 29.00 |
| Vocoder | HiFi-GAN | 6.33 | 0.00 |
| | Vocos | 3.00 | 0.00 |
| | BigVGAN | 0.33 | 0.00 |
| Removal Attack | Overwriting | 6.33 | 1.33 |
| | Averaging - AudioSeal | 0.67 | 0.00 |
| | Averaging - FakeMark$^A$ | 0.67 | 0.00 |
| | Averaging - FakeMark$^T$ | 0.67 | 0.00 |

**FakeMark remains effective for source verification on unseen architectures, and its behavior aligns with its robustness to the applied distortions.** Both FakeMark variants achieve low EERs (below 1%) across many distortions. The distortions that lead to higher EERs (signal processing and neural codecs) follow degradation patterns reported earlier in Tables 1 and 2. This suggests that the underlying vulnerabilities for unseen deepfake systems stem from the distortions themselves rather than from generalization failure. Between the two variants, FakeMark$^T$ generally shows stronger robustness than its time-domain counterpart.

## 5 CONCLUSION

Motivated by the limitations of both classifier-based and watermarking-based solutions for deepfake speech attribution, we proposed a novel watermarking framework FakeMark to enhance deepfake traceability. The core novelty of FakeMark is the injection of artifact-correlated watermarks, which allows the decoder to leverage both watermarks and deepfake artifacts for attribution. Our results confirm that such design provides improved generalization and robustness across various seen and unseen datasets and under distortions.

**Limitations** of this work include balancing the trade-off between robustness and speech quality, which could be addressed by selectively controlling which regions of the speech signal are watermarked. We acknowledge that the current design may remain vulnerable to stronger or adaptive removal attacks. Addressing these threats remains an important direction for future work.

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

# A APPENDIX

## A.1 THE USE OF LARGE LANGUAGE MODELS

Large Language Models (LLMs) were used only to polish the final version of the manuscript. The authors confirm that their original intention and meaning were not altered during this process.

## A.2 LIST OF AUGMENTATIONS DURING TRAINING

Below is the list of transformations used as augmentation strategies during all system training in our experiments. They are reproduced from the AudioSeal (Roman et al., 2024) pipeline. During training, each transformation is selected at random with equal probability and the chosen transform is applied to the current mini-batch. They include:

1. EnCodec: Inputs are resamples to 24kHz, compressed and reconstructed with En-Codec (Défossez et al., 2023) with $nq = 16$, and resampled back to 16kHz.

2. Speed: Playback speed of input signal is changed randomly between 0.9 and 1.1.

3. Resample: Inputs are resampled to 32kHz and resampled back to 16kHz.

4. Echo: A delay and less loud copy of the original is added to the input signal. Delay time is randomly sampled between 0.1 and 0.5 seconds, volume of the copied signal is randomly chosen between 0.1 and 0.5.

5. White noise: Gaussian noise with standard deviation fixed at 0.001 is added to the input signals.

6. Pink noise: Pink noise with standard deviation fixed at 0.01 is added to the input signals.

7. Lowpass filtering: A lowpass filter is applied to the input signal with a cutoff frequency at 5kHz.

8. Highpass filtering: A highpass filter is applied to the input signal with a cutoff frequency at 500Hz.

9. Bandpass filtering: A bandpass filter is applied to the input signal with a lower cutoff frequency of 300Hz and an upper cutoff frequency of 8kHz.

10. Smoothing: Inputs are smoothed using a moving average filter with a variable window size between 2 and 10.

11. Boost: Amplitude of input signal is multiplied by 1.2.

12. Duck: Amplitude of input signal is multiplied by 0.8.

13. AAC: Input signal is encoded in AAC format at 128kbps bitrate.

14. MP3: Input signal is encoded in MP3 format at 128kbps bitrate.

15. Identity: Returns the unprocessed input signal.

## A.3 FAKEMARK MODULE ARCHITECTURES

**FakeMark**[A]  We adopt the original AudioSeal generator architecture. The encoder uses a 1D convolution (32 channels, kernel size 7) followed by four convolutional blocks, each containing a residual unit (two kernel-3 convolutions with skip connection, doubling channels during down-sampling) and a down-sampling convolution (stride $S$, kernel $K = 2S$; $S = 2, 4, 5, 8$). It concludes with a two-layer LSTM and a final 1D convolution (128 channels, kernel 7) using ELU activations. The decoder mirrors the encoder with transposed convolutions and reversed strides. The latent dimension $H$ is 128.

**FakeMark**[T]  We adopt the Timbre encoder architecture but with larger size. A 1024-point Short-Time Fourier Transform (STFT) with 256 hop length is applied to obtain the magnitude spectrogram and phase of the input signal. The magnitude is fed to the 4-layer Carrier Encoder to obtain the encoded carrier feature, which is then concatenated with the original magnitude and the repeated watermark embedding $\boldsymbol{E}_w$. This combined feature is passed to the 5-layer Watermark Embedder to generate the magnitude spectrogram of watermark signal. The watermarked magnitude spectrogram

is obtained by adding watermark magnitude with original clean magnitude. This is different to the original Timber implementation where the Watermark Embedder directly outputs the watermarked magnitude. The watermarked signal is reconstructed via inverse STFT using the original phase and watermarked magnitude. The same original phase is also used for generating watermark waveform with watermark magnitude. The latent dimension $H$ is 513.

**Decoder**   We use an identical decoder architecture for both FakeMark generators. The decoder contains a pre-trained wav2vec model (namely the MMS-300M) as front-end. It extract a 1024-dimensional sequence-level representations from the input signal. These representations are then passed through a global average pooling layer to aggregate temporal information, followed by a fully connected layer that produced the output probabilities of 12 classes.

A.4   DATASETS DETAILS

Both the MLAAD_v5 dataset and source tracing challenge protocol can be downloaded from `https://deepfake-total.com/sourcetracing`. Dataset summary is presented in Table 6.

The ASVspoof5 dataset can be downloaded from `https://huggingface.co/datasets/jungjee/asvspoof5`. The TIMIT-TTS dataset can be downloaded from `https://zenodo.org/records/6560159`. They are summarized in Table 7.

Table 6: Summary of TTS models, Class ID, watermark bits, and number of samples in train, validation, and test sets of MLAAD_v5 dataset.

| TTS Model | Class ID | Watermark Bits | Train | Validation | Test |
|---|---|---|---|---|---|
| Mars5 | 0 | (0,1,0,0) | 275 | 23 | 300 |
| MeloTTS | 1 | (0,0,1,0) | 274 | 22 | 300 |
| Metavoice-1B | 2 | (1,1,1,0) | 267 | 29 | 300 |
| facebook-mms-tts-deu | 3 | (1,1,0,0) | 265 | 31 | 300 |
| tts_models-en-ljspeech-fast_pitch | 4 | (1,0,1,1) | 277 | 23 | 0 |
| tts_models-it-mai_female-glow-tts | 5 | (1,0,1,0) | 277 | 18 | 0 |
| griffin_lim | 6 | (0,1,1,1) | 1359 | 125 | 300 |
| suno-bark | 7 | (0,0,0,1) | 137 | 16 | 79 |
| suno-bark-small | 7 | (0,0,0,1) | 126 | 19 | 221 |
| tts_models-en-ljspeech-tacotron2-DCA | 8 | (1,1,1,1) | 272 | 25 | 49 |
| tts_models-fr-mai-tacotron2-DDC | 8 | (1,1,1,1) | 264 | 34 | 65 |
| tts_models-de-thorsten-tacotron2-DDC | 8 | (1,1,1,1) | 261 | 36 | 64 |
| tts_models-en-ljspeech-tacotron2-DDC | 8 | (1,1,1,1) | 142 | 11 | 32 |
| tts_models-en-ljspeech-tacotron2-DDC_ph | 8 | (1,1,1,1) | 135 | 11 | 90 |
| tts_models-en-ljspeech-speedy-speech | 9 | (1,0,0,0) | 268 | 28 | 0 |
| tts_models-it-mai_male-vits | 10 | (0,0,1,1) | 272 | 26 | 44 |
| tts_models-fr-css10-vits | 10 | (0,0,1,1) | 270 | 27 | 62 |
| tts_models-it-mai_female-vits | 10 | (0,0,1,1) | 269 | 29 | 60 |
| tts_models-lt-cv-vits | 10 | (0,0,1,1) | 264 | 34 | 53 |
| tts_models-de-css10-vits-neon | 10 | (0,0,1,1) | 264 | 35 | 60 |
| tts_models-en-ljspeech-vits--neon | 10 | (0,0,1,1) | 261 | 37 | 21 |
| tts_models-multilingual-multi-dataset-xtts_v2 | 11 | (1,1,0,1) | 1898 | 185 | 154 |
| tts_models-multilingual-multi-dataset-xtts_v1.1 | 11 | (1,1,0,1) | 1623 | 157 | 128 |
| vixTTS | 11 | (1,1,0,1) | 280 | 19 | 18 |

A.5   TRAINING AND IMPLEMENTATION DETAILS

**AudioSeal**   We use the official AudioSeal implementation from `https://github.com/facebookresearch/audioseal`.

**Timbre**   We use the official Timbre implementation from `https://github.com/TimbreWatermarking/TimbreWatermarking`.

**FakeMark**   For FakeMark training, the learning rate was linearly increased to $1 \times 10^{-4}$ over the first 2,000 mini-batches, and then linearly decayed to 0 at the 50,000th mini-batch, where training

Table 7: TTS models, source dataset, Class IDs, watermark bits, and sample counts for cross-dataset evaluation.

| TTS Model | Source Dataset | Class ID | Watermark Bits | Number of Samples |
|---|---|---|---|---|
| A01-GlowTTS | ASVspoof5 | 5 | (1,0,1,0) | 160 |
| A07-FastPitch | ASVspoof5 | 4 | (1,0,1,1) | 160 |
| fastpitch | TIMIT-TTS | 4 | (1,0,1,1) | 160 |
| glowtts | TIMIT-TTS | 5 | (1,0,1,0) | 160 |
| A11-Tacotron2 | ASVspoof5 | 8 | (1,1,1,1) | 160 |
| A29-XTTS | ASVspoof5 | 11 | (1,1,0,1) | 160 |
| A08-VITS | ASVspoof5 | 10 | (0,0,1,1) | 137 |
| vits | TIMIT-TTS | 10 | (0,0,1,1) | 23 |

stops. All input signals were resampled to 16 kHz if necessary. The waveform amplitude of training samples was randomly adjusted according to the Active Speech Level (ASL) based on ITU-T P.56. Training data were dynamically sampled by grouping files of similar durations and zero-padding them to form mini-batches, with a maximum batch duration of 40 seconds. Files longer than 10 seconds were randomly trimmed to durations between 6 and 10 seconds during training.

Validation was performed every 500 mini-batches, and the best model was selected based on the lowest sum of attribution loss and watermark extraction loss. Test samples were neither amplitude-adjusted nor trimmed.

The balancing weights for training were set as follows: attribution loss, 10.0; watermark extraction loss, 10.0; HiFi-GAN losses, 1.0 for FakeMark$^A$ and 7.0 for FakeMark$^T$ (with $L_1$ spectrogram loss weight 1.0 and feature matching loss weight 1.0); AudioSeal perceptual losses, 0.1 for $L_1$ loss, 10.0 for loudness loss, and 1.0 for frequency magnitude loss.

AudioSeal was trained on 6 NVIDIA A100 GPUs. The left training were performed on a single NVIDIA H100 GPU.

**MMS-300M Classifier** We adopt the same architecture as the FakeMark decoder and use the same codebase and training procedure, except that the maximum learning rate is set to $1 \times 10^{-5}$ and the batch size is fixed at 16. Training stops after 30,000 mini-batches. Best model selection is based on the classification accuracy on validation set.

**ResNet34 Classifier** We use a standard ResNet34 architecture with a temporal statistics pooling layer (TSPL) to extract a 128-dimensional embedding from the input signal, followed by a fully connected layer for prediction. The input is a randomly selected 4-second segment of the original signal, padded if shorter. Following Klein et al. (2025), we use 80-dimensional log linear filter-bank (LFB) features of the speech signal, computed with a 400-sample window, 160-sample hop, and a 400-point FFT. We further compute delta ($\Delta$) and double-delta ($\Delta\Delta$) features, and apply cepstral mean and variance normalization (CMVN), yielding a final feature dimension of 240. The model is trained using the Large Margin Cosine Loss with default settings from the implementation in `https://github.com/YirongMao/softmax_variants/blob/master/model_utils.py#L103`. All hyperparameters are identical to those used for MMS-300M training, except that the maximum learning rate is set to $1 \times 10^{-4}$.

A.6 LIST OF DISTORTIONS DURING EVALUATION

The settings of distortion and watermark removal attacks are:

1. Pitch shift: Pitch is randomly shifted between -1 and 1 semitones.

2. Playback speed: Original speed is adjust to a number randomly sampled between 0.95 and 1.05.

3. Noise: Random noise from MUSAN noise recordings is applied at 0dB SNR.

4. BigVGAN: Using code and pre-trained weight from `https://github.com/NVIDIA/BigVGAN`. Input signals are resampled to 24kHz, passed to BigVGAN vocoder, and resampled back to 16kHz.

5. HiFi-GAN: Using the pre-trained weights from `https://huggingface.co/speechbrain/tts-hifigan-libritts-16kHz`.

6. Vocos: Using code and pre-trained weight (vocos-mel-24khz) from `https://github.com/gemelo-ai/vocos/tree/main`. Input signals are resampled to 24kHz, passed to Vocos, and resampled back to 16kHz.

7. SpeechTokenizer: Using code and pre-trained weight (speechtokenizer_hubert_avg) from `https://github.com/ZhangXInFD/SpeechTokenizer`.

8. FACodec: Using code and pre-trained weight from `https://huggingface.co/amphion/naturalspeech3_facodec`.

9. WavTokenizer: Using code and pre-trained weight (WavTokenizer-small-600-24k-4096) from `https://huggingface.co/amphion/naturalspeech3_facodec`. Input signals are resampled to 24kHz, passed to WavTokenizer, and resampled back to 16kHz.

10. Overwriting: Input signals are passed sequentially through a pre-trained WavMark model, resampled to 44.1 kHz, processed to a pre-trained SilentCipher model, and then resampled back to 16 kHz. This sequence is repeated three times to obtain the watermarked signal.

11. Averaging: Data samples from the zh-CN subset of the Common Voice dataset are processed using the evaluated AudioSeal, FakeMark$^A$, and FakeMark$^T$ models. The resulting watermark signals for each sample are summed and averaged, and this averaged watermark is then subtracted from the input signal.

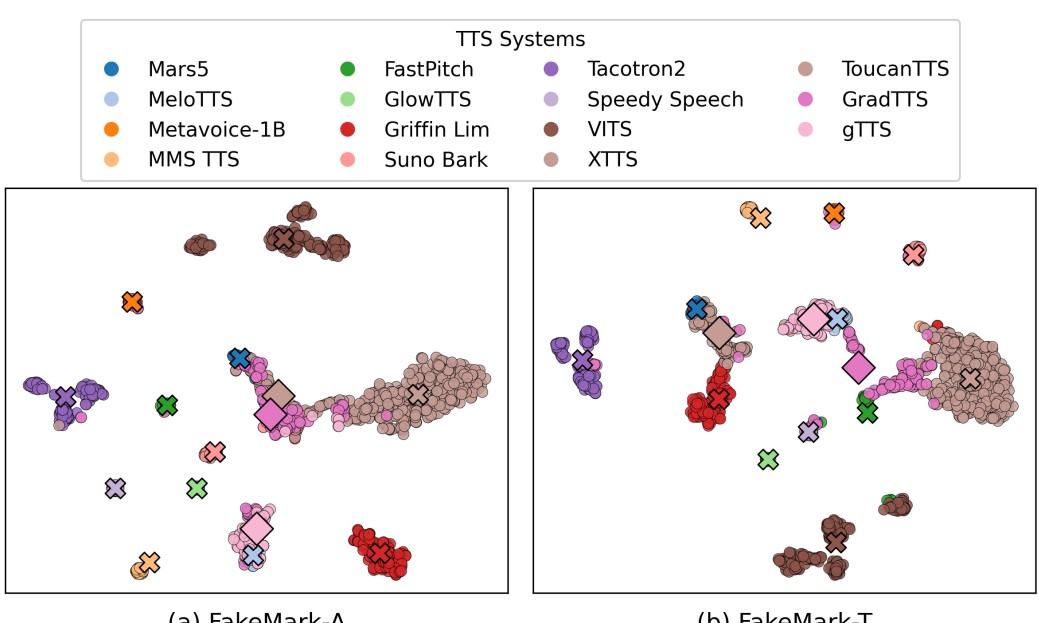

(a) FakeMark-A         (b) FakeMark-T

Figure 3: t-SNE visualization of clean-speech embeddings from the MLAAD_v5 validation set and the ASVspoof5 + TIMIT-TTS unseen set. Each point corresponds to a clean utterance generated by a particular TTS system, and the × markers denote the centroid of seen training system, whereas the ■ markers denote the centroid of unseen test system.

## A.7 DETAILS OF CLOSEST SYSTEM MAPPING

### A.7.1 DESCRIPTION

To determine the closest training system for each unseen system, we compute embedding centroid (visualized in Figure 3) using the clean, unwatermarked MLAAD_v5 validation set. The embedding

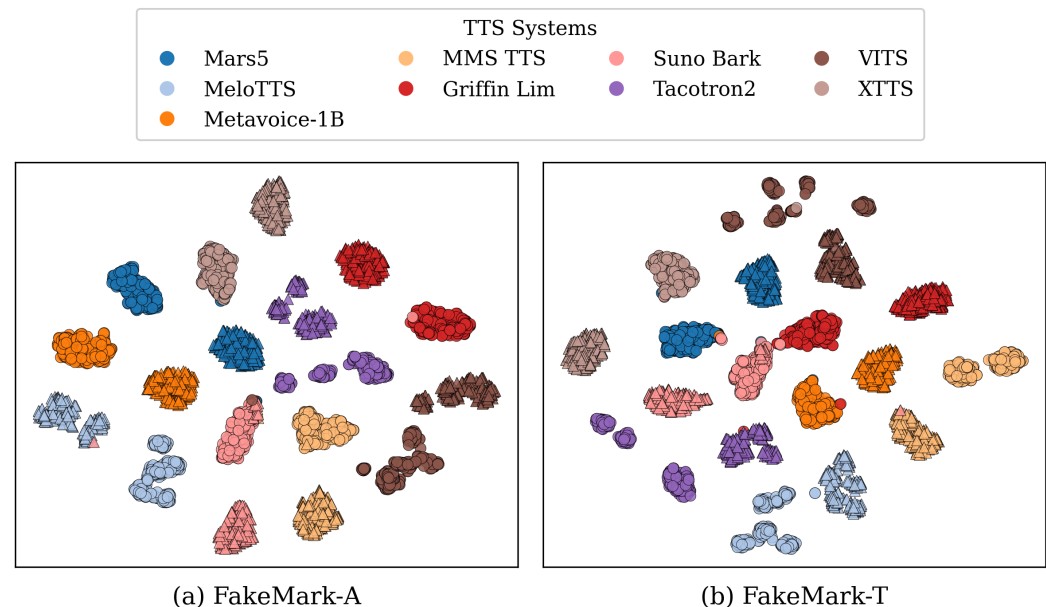

Figure 4: t-SNE visualization of embeddings from the MLAAD_v5 test set. Circle markers (●) correspond to embeddings from clean, unwatermarked inputs, whereas triangle markers (▲) correspond to embeddings from watermarked inputs.

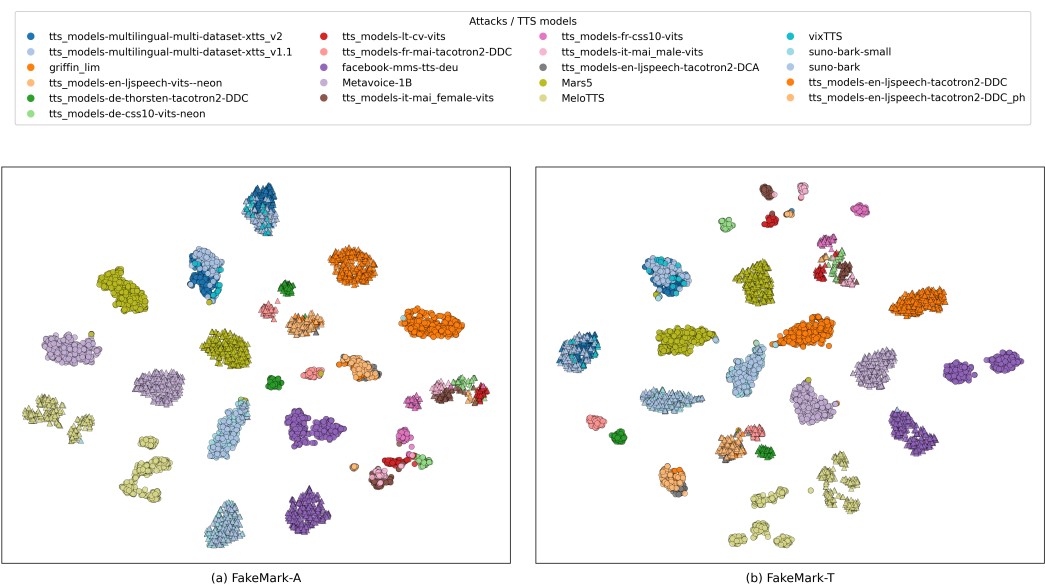

Figure 5: t-SNE visualization of the same embedding representations as in Figure 4. TTS systems are colored by their original labels (24 classes), whereas in Figure 4 they are colored by combined labels (12 classes). Circle markers (●) correspond to embeddings from clean, unwatermarked waveforms, whereas triangle markers (▲) correspond to embeddings from watermarked inputs.

of a signal is the output sequence-level representations of MMS-300M after being averaged along time axis. Each system centroid is obtained by averaging embeddings extracted from that system. We then compute the pairwise cosine similarity between all system centroids, and define the closest-system mapping as the system with the highest cosine similarity. The resulting mappings for both FakeMark variants are reported in Table 8. Dataset details are reported in Table 9, they are used to generate 300 random pairs of reference and test signals for performance evaluation.

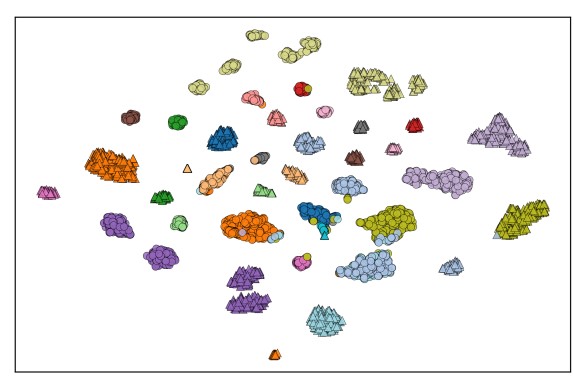
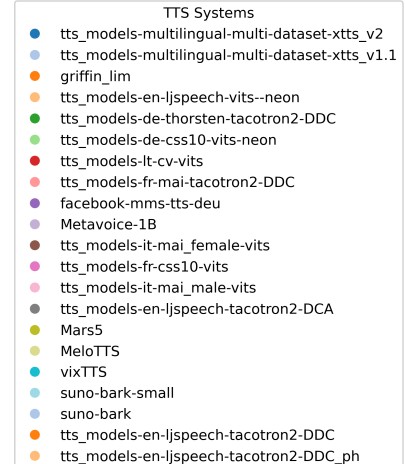

Figure 6: t-SNE visualization of embedding representations extracted by the FakeMark$^A$ decoder using MLAAD_v5 test set. It is trained using original 24-class labels of the training set, instead of the merged 12-class labels. Circle markers (●) correspond to embeddings from clean, unwatermarked waveforms, whereas triangle markers (▲) correspond to embeddings from watermarked inputs.

### A.7.2   ADDITIONAL VISUALIZATIONS

We provide additional analysis of the embedding representations on MLAAD_v5 test set in Figures 4 (colored by 12 training classes) and 5 (colored by original 24 classes). Figure 6 shows embedding representations on MLAAD_v5 test set when FakeMark$^A$ is trained using the original 24-class training labels rather than the merged 12-class labels.

We notice that most TTS systems form clear and distinct clusters. For example, in Figures 5, the VITS variants cluster in close proximity. Similarly, xtts_v2, xtts_v1.1, and vixTTS (a fine-tuned version of xtts_v2) exhibit overlapping clusters. However, in Figure 6, systems with different architectures may also appear closer in the embedding space, leading to partial overlap.

### A.8   ADDITIONAL RESULTS ON CHALLENGING SEQUENTIAL DISTORTIONS

We present additional attribution results under sequential distortions, where a reconstruction-based neural codec or vocoder is applied first, followed by a second distortion consisting of signal transformations or adversarial attacks. Results for in-domain evaluation are shown in Table 10, and cross-dataset results are shown in Table 11.

Across all second distortions, the evaluated models exhibit performance trends consistent with their original results in Tables 1 and 2. For the first-stage distortions, a similar pattern is observed, with the exception that all models degrade more severely when WavTokenizer is applied. This outcome aligned with earlier observations in Tables 1 and 2. The combination of WavTokenizer and MUSAN noise produces the lowest attribution accuracy across all models, as both distortions are individually among the strongest stressors.

The two classifier baselines remain robust on the in-domain MLAAD_v5 test set but experience substantial performance drops on cross-dataset data. In contrast, Timbre and the proposed FakeMark$^T$ achieve reliable performance on both evaluation sets, demonstrating stronger cross-dataset resilience.

### A.9   VISUALIZATIONS OF SPEECH SIGNALS

Table 8: Full cosine similarities between unseen TTS systems and all training systems in the embedding spaces of FakeMark$^A$ and FakeMark$^T$.

| Training System | FakeMark$^A$ | FakeMark$^T$ |
|---|---|---|
| **ToucanTTS** | | |
| Mars5 | **0.686** | **0.893** |
| MeloTTS | 0.039 | 0.199 |
| Metavoice-1B | -0.004 | 0.049 |
| MMS TTS | -0.102 | -0.021 |
| FastPitch | -0.156 | -0.369 |
| GlowTTS | -0.221 | -0.198 |
| Griffin Lim | -0.112 | 0.481 |
| Suno Bark | 0.027 | -0.018 |
| Tacotron2 | -0.120 | -0.048 |
| Speedy Speech | -0.132 | -0.086 |
| VITS | -0.129 | -0.227 |
| XTTS | 0.538 | -0.033 |
| **GradTTS** | | |
| Mars5 | 0.541 | -0.056 |
| MeloTTS | 0.293 | 0.236 |
| Metavoice-1B | -0.077 | -0.136 |
| MMS TTS | -0.102 | -0.166 |
| FastPitch | -0.088 | 0.064 |
| GlowTTS | -0.272 | -0.216 |
| Griffin Lim | -0.148 | -0.111 |
| Suno Bark | -0.018 | 0.031 |
| Tacotron2 | -0.015 | 0.004 |
| Speedy Speech | -0.076 | 0.068 |
| VITS | -0.105 | -0.210 |
| XTTS | **0.550** | **0.831** |
| **gTTS** | | |
| Mars5 | -0.027 | 0.060 |
| MeloTTS | **0.982** | **0.985** |
| Metavoice-1B | 0.004 | 0.014 |
| MMS TTS | -0.079 | -0.039 |
| FastPitch | -0.162 | -0.229 |
| GlowTTS | -0.122 | -0.096 |
| Griffin Lim | -0.133 | 0.171 |
| Suno Bark | 0.116 | -0.107 |
| Tacotron2 | -0.069 | -0.165 |
| Speedy Speech | -0.141 | -0.150 |
| VITS | -0.072 | -0.179 |
| XTTS | 0.068 | -0.014 |

Table 9: TTS models, source dataset, closest training system, and sample counts for unseen cross-dataset evaluation.

| TTS Model | Source Dataset | Closest Training System | Number of Samples |
|---|---|---|---|
| ToucanTTS | ASVspoof5 | Mars5 | 100 |
| GradTTS | ASVspoof5 | XTTS | 100 |
| gTTS | TIMIT-TTS | MeloTTS | 100 |

Table 10: Attribution accuracy results under two stages of distortion for MLAAD_v5 test set: (1) neural codec and vocoder regeneration, and (2) signal transforms and adversarial attacks.

| Distortion1 | Distortion2 | Proposed Method | | Watermarking Baselines | | Classifier Baselines | |
|---|---|---|---|---|---|---|---|
| | | FakeMark$^A$ | FakeMark$^T$ | AudioSeal | Timbre | MMS-300M | ResNet34 |
| BigVGAN | Averaging - AudioSeal | 0.98 | 1.00 | 0.24 | 1.00 | 0.98 | 1.00 |
| | Averaging - FakeMark$^A$ | 0.95 | 1.00 | 0.15 | 1.00 | 0.98 | 1.00 |
| | Averaging - FakeMark$^T$ | 0.98 | 1.00 | 0.21 | 1.00 | 0.97 | 0.98 |
| | Noise | 0.41 | 0.75 | 0.16 | 0.54 | 0.78 | 0.46 |
| | Overwriting | 0.94 | 0.62 | 0.17 | 1.00 | 0.92 | 0.54 |
| | Pitch | 0.84 | 0.94 | 0.20 | 0.96 | 0.16 | 0.87 |
| | Speed | 0.98 | 0.97 | 0.25 | 0.97 | 0.97 | 0.98 |
| FACodec | Averaging - AudioSeal | 0.95 | 0.89 | 0.13 | 0.83 | 0.94 | 0.79 |
| | Averaging - FakeMark$^A$ | 0.92 | 0.89 | 0.08 | 0.80 | 0.92 | 0.78 |
| | Averaging - FakeMark$^T$ | 0.92 | 0.90 | 0.18 | 0.83 | 0.95 | 0.73 |
| | Noise | 0.44 | 0.63 | 0.13 | 0.25 | 0.68 | 0.30 |
| | Overwriting | 0.67 | 0.49 | 0.12 | 0.60 | 0.84 | 0.32 |
| | Pitch | 0.62 | 0.79 | 0.15 | 0.75 | 0.11 | 0.75 |
| | Speed | 0.94 | 0.84 | 0.17 | 0.80 | 0.92 | 0.89 |
| HiFi-GAN | Averaging - AudioSeal | 0.92 | 1.00 | 0.12 | 1.00 | 0.90 | 0.97 |
| | Averaging - FakeMark$^A$ | 0.89 | 1.00 | 0.05 | 1.00 | 0.92 | 0.97 |
| | Averaging - FakeMark$^T$ | 0.95 | 1.00 | 0.08 | 1.00 | 0.92 | 0.95 |
| | Noise | 0.41 | 0.67 | 0.10 | 0.53 | 0.56 | 0.41 |
| | Overwriting | 0.79 | 0.75 | 0.11 | 0.99 | 0.84 | 0.59 |
| | Pitch | 0.78 | 0.95 | 0.10 | 0.97 | 0.14 | 0.87 |
| | Speed | 0.95 | 1.00 | 0.10 | 0.98 | 0.92 | 0.94 |
| SpeechTokenizer | Averaging - AudioSeal | 0.86 | 0.97 | 0.09 | 0.92 | 0.87 | 0.94 |
| | Averaging - FakeMark$^A$ | 0.83 | 0.97 | 0.06 | 0.93 | 0.86 | 0.94 |
| | Averaging - FakeMark$^T$ | 0.84 | 0.97 | 0.09 | 0.92 | 0.86 | 0.87 |
| | Noise | 0.37 | 0.65 | 0.10 | 0.30 | 0.63 | 0.40 |
| | Overwriting | 0.65 | 0.57 | 0.09 | 0.65 | 0.76 | 0.54 |
| | Pitch | 0.59 | 0.98 | 0.10 | 0.86 | 0.14 | 0.92 |
| | Speed | 0.89 | 0.92 | 0.11 | 0.90 | 0.89 | 0.89 |
| Vocos | Averaging - AudioSeal | 1.00 | 1.00 | 0.10 | 1.00 | 0.97 | 1.00 |
| | Averaging - FakeMark$^A$ | 0.97 | 1.00 | 0.06 | 1.00 | 0.98 | 1.00 |
| | Averaging - FakeMark$^T$ | 0.97 | 1.00 | 0.15 | 1.00 | 0.90 | 1.00 |
| | Noise | 0.51 | 0.57 | 0.11 | 0.55 | 0.57 | 0.44 |
| | Overwriting | 0.81 | 0.57 | 0.10 | 0.99 | 0.84 | 0.54 |
| | Pitch | 0.83 | 0.94 | 0.14 | 0.95 | 0.16 | 0.92 |
| | Speed | 0.98 | 0.97 | 0.15 | 0.97 | 0.95 | 0.94 |
| WavTokenizer | Averaging - AudioSeal | 0.35 | 0.30 | 0.08 | 0.20 | 0.44 | 0.59 |
| | Averaging - FakeMark$^A$ | 0.33 | 0.30 | 0.06 | 0.18 | 0.38 | 0.62 |
| | Averaging - FakeMark$^T$ | 0.37 | 0.33 | 0.10 | 0.19 | 0.41 | 0.67 |
| | Noise | 0.21 | 0.19 | 0.10 | 0.09 | 0.22 | 0.22 |
| | Overwriting | 0.38 | 0.22 | 0.10 | 0.16 | 0.37 | 0.22 |
| | Pitch | 0.10 | 0.32 | 0.09 | 0.20 | 0.06 | 0.57 |
| | Speed | 0.30 | 0.35 | 0.08 | 0.19 | 0.41 | 0.63 |

Table 11: Attribution accuracy results under two stages of distortion for ASVspoof5 and TIMIT-TTS datasets: (1) neural codec and vocoder regeneration, and (2) signal transforms and adversarial attacks.

| Distortion1 | Distortion2 | Proposed Method | | Watermarking Baselines | | Classifier Baselines | |
|---|---|---|---|---|---|---|---|
| | | FakeMark$^A$ | FakeMark$^T$ | AudioSeal | Timbre | MMS-300M | ResNet34 |
| BigVGAN | Averaging - AudioSeal | 1.00 | 1.00 | 0.13 | 1.00 | 0.08 | 0.21 |
| | Averaging - FakeMark$^A$ | 1.00 | 1.00 | 0.20 | 1.00 | 0.08 | 0.21 |
| | Averaging - FakeMark$^T$ | 0.96 | 1.00 | 0.17 | 1.00 | 0.08 | 0.21 |
| | Noise | 0.29 | 0.58 | 0.07 | 0.58 | 0.04 | 0.08 |
| | Overwriting | 0.88 | 0.33 | 0.12 | 0.92 | 0.08 | 0.04 |
| | Pitch | 0.79 | 0.83 | 0.12 | 0.96 | 0.00 | 0.17 |
| | Speed | 0.96 | 0.79 | 0.11 | 0.97 | 0.08 | 0.21 |
| FACodec | Averaging - AudioSeal | 0.75 | 0.25 | 0.03 | 0.78 | 0.12 | 0.04 |
| | Averaging - FakeMark$^A$ | 0.71 | 0.29 | 0.13 | 0.84 | 0.08 | 0.04 |
| | Averaging - FakeMark$^T$ | 0.62 | 0.33 | 0.08 | 0.85 | 0.08 | 0.08 |
| | Noise | 0.21 | 0.21 | 0.05 | 0.31 | 0.08 | 0.00 |
| | Overwriting | 0.38 | 0.04 | 0.07 | 0.50 | 0.08 | 0.00 |
| | Pitch | 0.42 | 0.46 | 0.06 | 0.80 | 0.00 | 0.08 |
| | Speed | 0.67 | 0.33 | 0.07 | 0.80 | 0.12 | 0.08 |
| HiFi-GAN | Averaging - AudioSeal | 0.71 | 1.00 | 0.09 | 1.00 | 0.17 | 0.17 |
| | Averaging - FakeMark$^A$ | 0.71 | 1.00 | 0.05 | 1.00 | 0.17 | 0.17 |
| | Averaging - FakeMark$^T$ | 0.75 | 1.00 | 0.06 | 1.00 | 0.17 | 0.17 |
| | Noise | 0.21 | 0.58 | 0.05 | 0.55 | 0.08 | 0.08 |
| | Overwriting | 0.46 | 0.29 | 0.10 | 0.89 | 0.08 | 0.04 |
| | Pitch | 0.62 | 0.92 | 0.08 | 0.98 | 0.00 | 0.21 |
| | Speed | 0.79 | 1.00 | 0.07 | 0.98 | 0.17 | 0.17 |
| SpeechTokenizer | Averaging - AudioSeal | 0.50 | 1.00 | 0.04 | 0.85 | 0.08 | 0.08 |
| | Averaging - FakeMark$^A$ | 0.54 | 1.00 | 0.06 | 0.91 | 0.08 | 0.08 |
| | Averaging - FakeMark$^T$ | 0.58 | 1.00 | 0.04 | 0.90 | 0.08 | 0.08 |
| | Noise | 0.25 | 0.54 | 0.04 | 0.32 | 0.08 | 0.12 |
| | Overwriting | 0.42 | 0.17 | 0.10 | 0.49 | 0.08 | 0.00 |
| | Pitch | 0.21 | 0.92 | 0.07 | 0.85 | 0.00 | 0.12 |
| | Speed | 0.46 | 0.83 | 0.05 | 0.87 | 0.08 | 0.17 |
| Vocos | Averaging - AudioSeal | 1.00 | 0.96 | 0.05 | 1.00 | 0.08 | 0.21 |
| | Averaging - FakeMark$^A$ | 0.92 | 0.96 | 0.10 | 1.00 | 0.04 | 0.21 |
| | Averaging - FakeMark$^T$ | 0.96 | 0.96 | 0.07 | 1.00 | 0.08 | 0.17 |
| | Noise | 0.29 | 0.46 | 0.05 | 0.56 | 0.08 | 0.08 |
| | Overwriting | 0.62 | 0.29 | 0.07 | 0.91 | 0.04 | 0.08 |
| | Pitch | 0.71 | 0.88 | 0.06 | 0.96 | 0.00 | 0.12 |
| | Speed | 0.92 | 0.96 | 0.08 | 0.98 | 0.08 | 0.17 |
| WavTokenizer | Averaging - AudioSeal | 0.04 | 0.04 | 0.04 | 0.18 | 0.08 | 0.04 |
| | Averaging - FakeMark$^A$ | 0.04 | 0.04 | 0.08 | 0.20 | 0.08 | 0.04 |
| | Averaging - FakeMark$^T$ | 0.04 | 0.04 | 0.03 | 0.19 | 0.08 | 0.04 |
| | Noise | 0.00 | 0.04 | 0.04 | 0.10 | 0.00 | 0.00 |
| | Overwriting | 0.08 | 0.08 | 0.07 | 0.08 | 0.04 | 0.00 |
| | Pitch | 0.00 | 0.00 | 0.06 | 0.22 | 0.00 | 0.08 |
| | Speed | 0.04 | 0.00 | 0.03 | 0.20 | 0.08 | 0.08 |

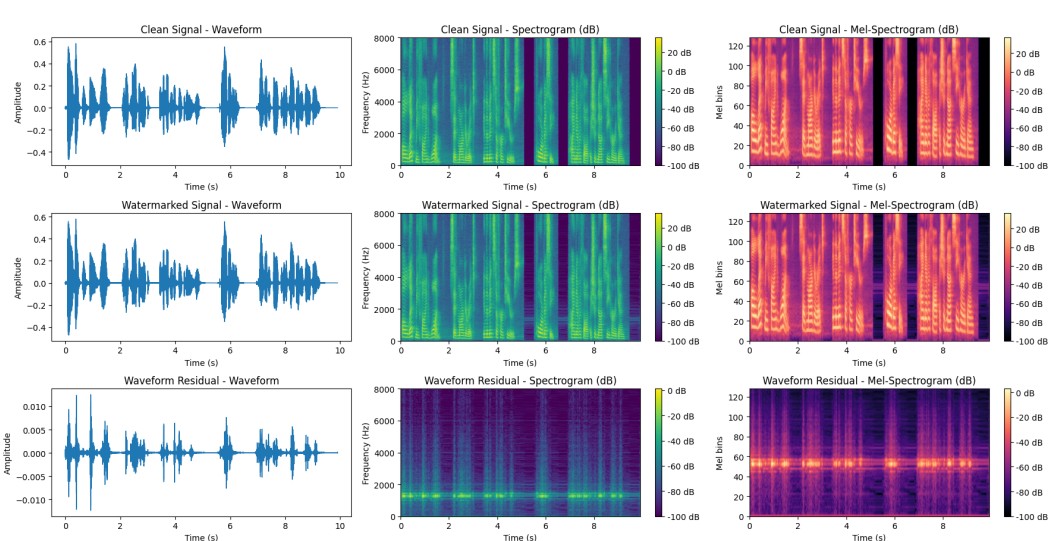

Figure 7: Visualization of AudioSeal watermarking on MLAAD-en-tts_models-en-ljspeech-tacotron2-DDC-northandsouth_27_f000104.

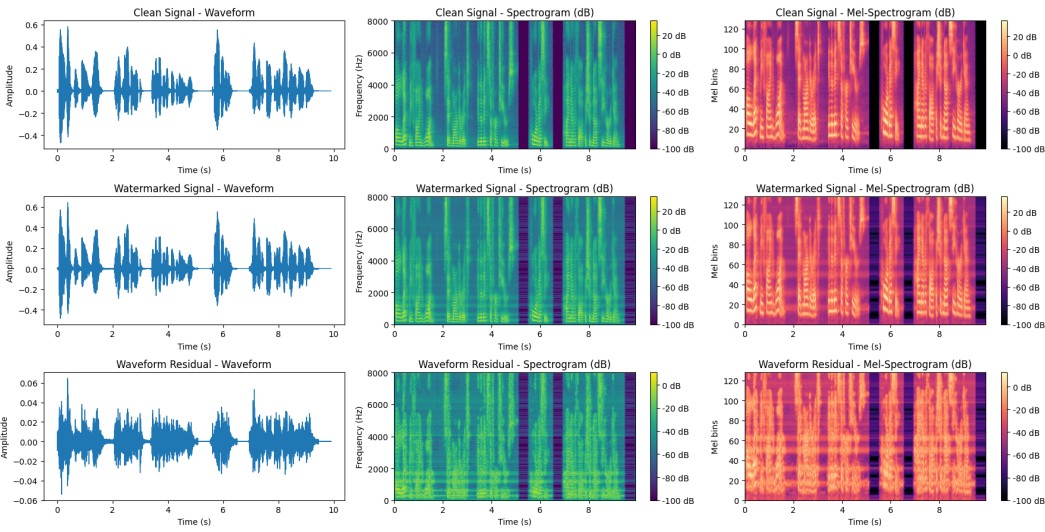

Figure 8: Visualization of Timbre watermarking on MLAAD-en-tts_models-en-ljspeech-tacotron2-DDC-northandsouth_27_f000104.

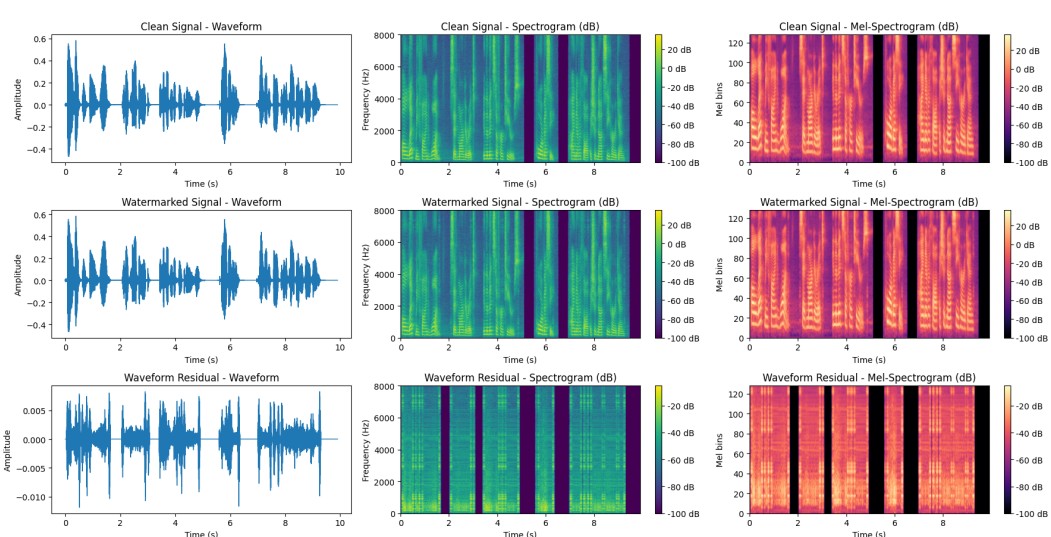

Figure 9: Visualization of FakeMark$^A$ watermarking on MLAAD-en-tts_models-en-ljspeech-tacotron2-DDC-northandsouth_27_f000104.

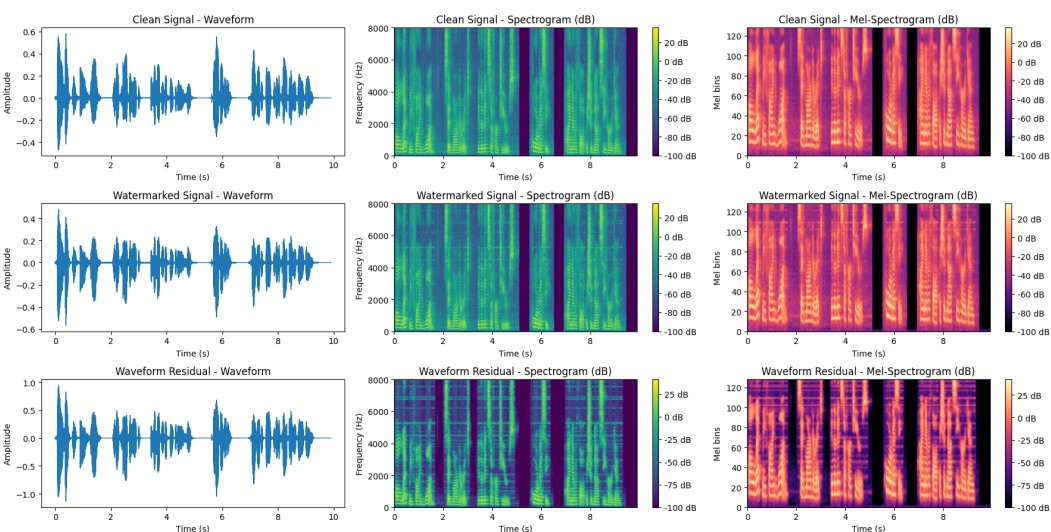

Figure 10: Visualization of FakeMark$^T$ watermarking on MLAAD-en-tts_models-en-ljspeech-tacotron2-DDC-northandsouth_27_f000104.

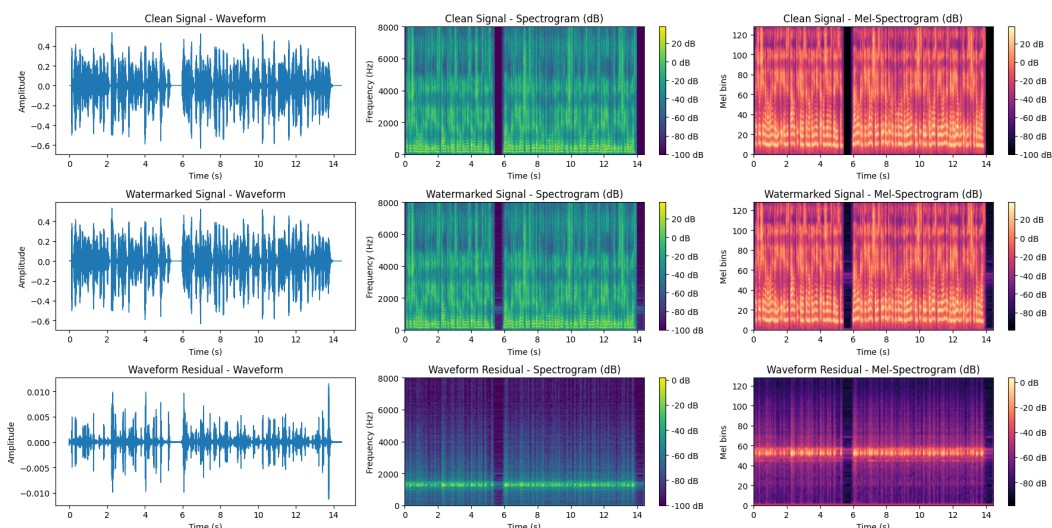

Figure 11: Visualization of AudioSeal watermarking on MLAAD-lt-tts_models-lt-cv-vits-emerald_city_of_oz_03_f000037.

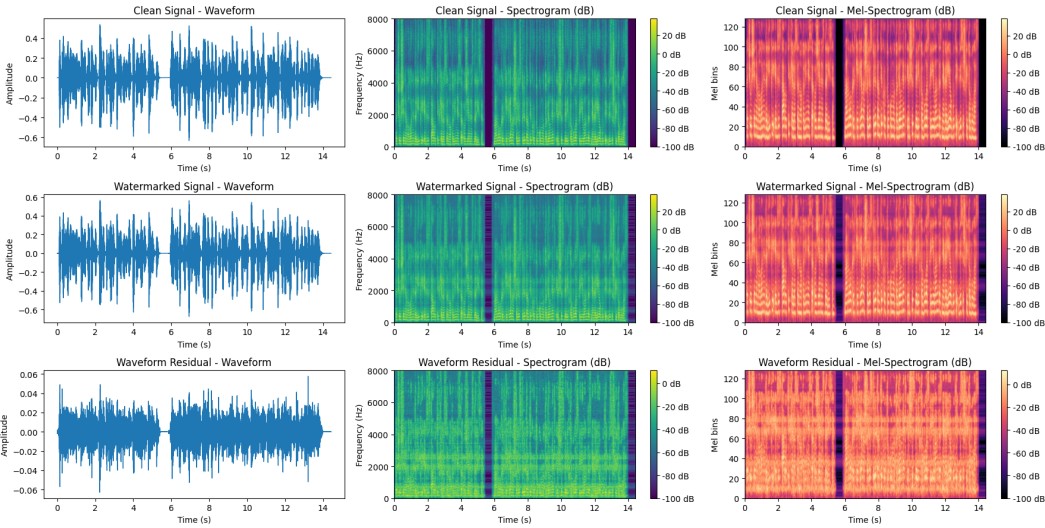

Figure 12: Visualization of Timbre watermarking on MLAAD-lt-tts_models-lt-cv-vits-emerald_city_of_oz_03_f000037.

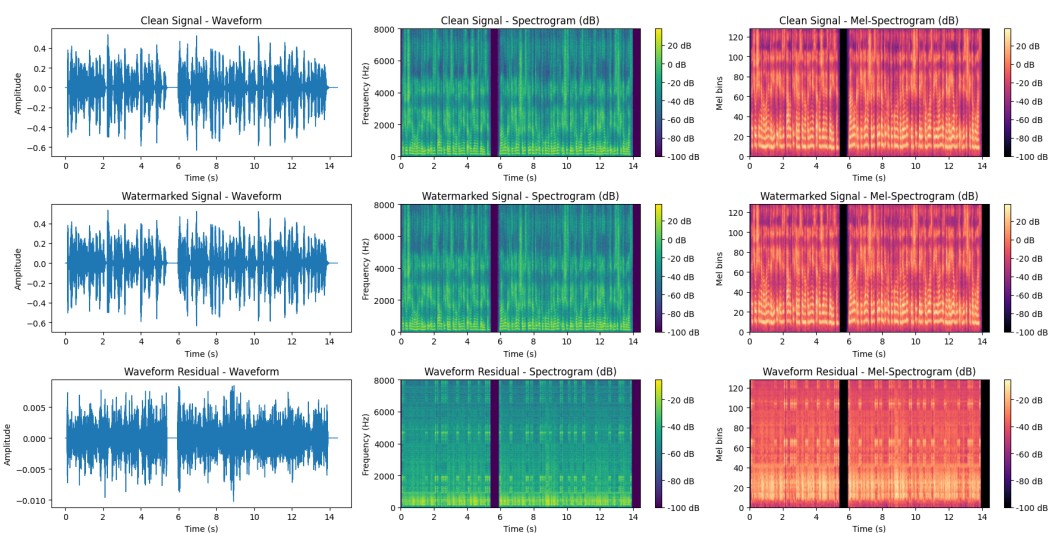

Figure 13: Visualization of FakeMark$^A$ watermarking on MLAAD-lt-tts_models-lt-cv-vits-emerald_city_of_oz_03_f000037.

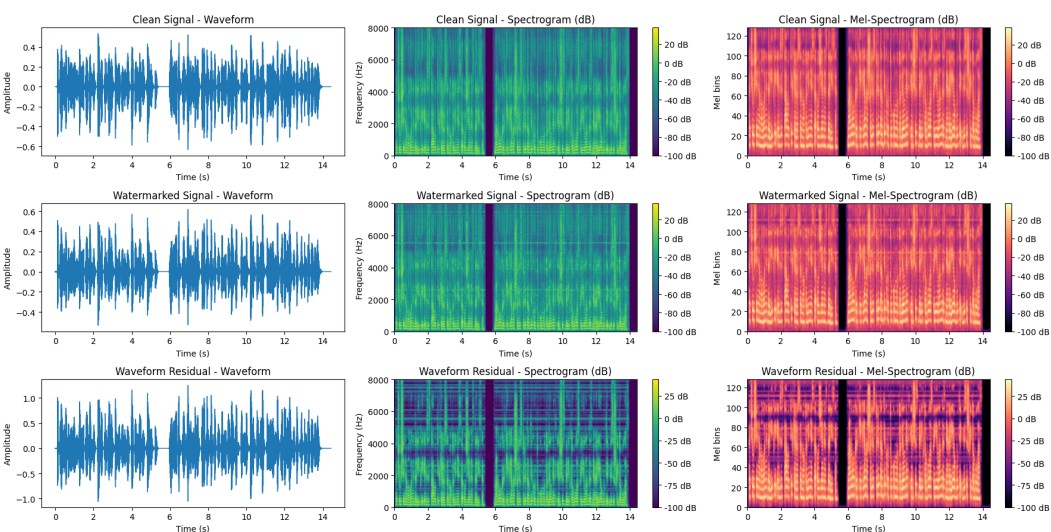

Figure 14: Visualization of FakeMark$^T$ watermarking on MLAAD-lt-tts_models-lt-cv-vits-emerald_city_of_oz_03_f000037.

