# OpenReview forum: "FakeMark: Deepfake Speech Attribution With Watermarked Artifacts"
_ICLR.cc/2026/Conference — Submitted to ICLR 2026_

### Official Review · Reviewer_z9Yp · 2025-10-23

**Soundness:** 3
**Presentation:** 2
**Contribution:** 3
**Rating:** 2
**Confidence:** 4

**Summary:**

This submission proposes a post-hoc watermarking technique dedicated to audio GenAI.
The idea is to make the watermark signal reinforcing the audio cues characterizing a particular GenAI model.
The application is data provenance.

**Strengths:**

The idea is novel. The only prior work that came to my mind is the paper *Learning to watermark LLM-Generated Text Via Reinforcement Learning*, Xu et al., 1st workshop on GenAI, where a passive detector is first trained to detect text generated by a particular LLM, and then this LLM is watermarked (weights fine-tuning) to better comply with this detector.

The same approach is applied here with noticeable differences: audio not text, post-hoc not in-gen watermarking, a classifier not a detector, learning of a specific watermark decoder not the original detector.

**Weaknesses:**

### W1 - Motivations in the introduction

- A first confusion in the introduction is about the final application. The terms *deepfake*, *detector* are misleading. For instance, the proposed scheme does not make the distinction between real audio and synthetic audio. It distinguishes the generated model knowing the audio is synthetic. Moreover, deepfake detection is an open-set problem, whereas the proposed scheme can only handle a fixed number $C$ of models. On the same token, speaking about copyright violation (Line 29) for deepfakes does not make sense. I would say that the term **data provenance** or **GenAI model identification** are missing. On the same token, I would replace the term *deepfake* by **synthetic** and *detector* by **classifier**.  Speech synthesis models can be very useful for other applications than deepfakes.  **This includes changing the title, the abstract, the introduction and even the nickname FakeMark.**

- I recommend to mitigate assertions like *watermarking-based solutions are easily compromised* (Line 13), *can be easily compromised by common distortions* (Line 41), *whereas watermark detectors degrade* (Line 48). Watermarking is a cat and mouse game. You are citing papers that succeed attacking some watermarking schemes. It does not mean all watermarking schemes are vulnerable to these attacks. For instance, there are watermarking schemes robust to neural codecs to some extend like *Latent Watermarking of Audio Generative Models*, San Roman et al. Moreover, Line 330, you admit that Timbre is indeed robust. One just has to add the neural codec in the augmentation set during training. Anyway, stating that watermarking is robust or not robust does not mean anything unless one looks at the distortion of the attack.

### W2 - lack of clarity
Some mathematical formalism would help understanding the training.
- Sect. 3.2
Line 211: *The back-propagated loss [...] implicitly guides the watermark embeddings*. This is technically wrong. Since that loss is computed over unwatermarked signals, it cannot impact the watermarked embedder.

- Sect. 4.1
Line 252: *evaluations on unseen architectures are beyond the scope of this work.* It is even more than that: Evaluations on unseen models is simply impossible since the watermarking scheme need the model for its training. This is a clear limitation of the proposal.

### W3 - lack of security
The watermark signal in the latent space is always constant, for a given models. Therefore the scheme is not secure. The attacker could average many latent vectors extracted from watermarked signals to estimate $\mathbf{e}_w$ and then remove it. Counter-measures are known for a long time: **Side-informed watermarking** would make $\mathbf{e}_w$ dependent on $\mathbf{e}_s$, ie. the signal to be watermarked. See the literature about **watermarking security**. The irony is that the paper *Can simple averaging defeat modern watermarks?*, Yang et al. is cited.

Another point is that the watermark decoder is based on an open-source SSL backbone. This is also a vulnerability, see
*Evaluation of security of ML-based watermarking: Copy and removal attacks*, Kinakh et al. WIFS 2024.

### W4 - Unfair benchmark
- Line 288: Overwriting as an attack? This is really unfair. AudioSeal and Timbre are *publicly-available* for research purpose. And I hope that your scheme will be as well. Therefore, either overwriting is non sense (and I strongly agree with this option), either you should also apply overwriting to your own scheme as well.

- Some measurements about AudioSeal surprise me. The SISNR reported in the original paper is low (26 dB - Table 1 / 24 dB Table 9) whereas your Table 3 reports 36dB. The original paper report almost perfect robustness against speed x1.25, whereas your Table 1 reads poor performance for speed x1.05, ie. a less harmful attack.

- The four techniques (FakeMark A and T, AudioSeal, Timbre) have strong differences in speech quality. I consider that a SISNR lower than 20dB is a dead-end. FakeMark^T is simply not useable. Moreover, comparing the robustness of watermarking scheme with much perceptibility differences is meaningless.

- As already mentioned, the *Averaging* attack is totally feasible against FakeMark. It does not operate in the time domain, but in the feature domain. According to the appendix, *Averaging* seems to target only AudioSeal. Therefore reporting the impact of this targeted attack on other schemes is weird. This comment also holds for *Overwritting*, especially against classifiers (last 2 columns in Table 1 and 2).

**Questions:**

### Q1

Line 216: Why is the message random? This means that the attribution loss and the watermarking detection loss disagree. A signal without watermark should be classified as generated by its ground-truth model, whereas the same signal with watermark should be classified in another way. How can the watermark signal  be aligned with generation artifacts in this case?

---

> ### Author Response · Authors · 2025-11-28
> **Response to Reviewer z9Yp (Part 1)**
>
> We thank the reviewer for their careful evaluation and valuable comments. Below, we address each of the points raised.
>
> ---
>
> **W1: Confusion on application of attribution systems**
>
> **We have revised lines 27–31 and Figure 1** to clarify that, similar to recent deepfake attribution and source-tracing literatures [1, 2], FakeMark is applied to audio data that are flagged as fake by a conventional deepfake detector. The deepfake detector is trained to distinguish real vs. fake audio, while attribution models like FakeMark focus specifically on attributing the generating model of the fake speech.
>
> [1] N. Klein, T. Chen, H. Tak, R. Casal, and E. Khoury, “Source Tracing of Audio Deepfake Systems,” in Proc. Interspeech, ISCA, Sept. 2024, pp. 1100–1104. doi: 10.21437/Interspeech.2024-1283.
>
> [2] Y. Xie et al., “Generalized Source Tracing: Detecting Novel Audio Deepfake Algorithm with Real Emphasis and Fake Dispersion Strategy,” in Interspeech 2024, 2024, pp. 4833–4837. doi: 10.21437/Interspeech.2024-254.
>
> ---
>
> **W1: Terminology on final use application**
>
> **We have modified the title, abstraction and introduction sections** as the reviewer suggests (line 27-31, line 13, 44, 49).
>
> We now adopt the term “system signature” as the primary term when describing the injected watermarks. This terminology aligns with the definitions in [3] where signatures refer to watermarks used to identify the owner or producer of the content, while fingerprints are watermarks used to identify individual buyers or consumers. Since FakeMark is designed to attribute the generative model rather than individual users, we believe that “signature” more accurately reflects its purpose than “fingerprint.”
>
> **We also clarified the intended use case** (lines 27–31, 88–93) to state that FakeMark does not aim to establish a cross-vendor global standard. Instead, it is intended for content-owner-controlled attribution, where the party generating or distributing synthetic audio injects its own system signature.
>
> [3] M. L. Miller et al., “A Review of Watermarking Principles and Practices,” in Digital Signal Processing in Multimedia Systems, ch. 18, pp. 461–485, 1999.
>
> ---
>
> **W2: lack of clarity**
>
> > Sect. 3.2 Line 211: The back-propagated loss [...] implicitly guides the watermark embeddings. This is technically wrong. Since that loss is computed over unwatermarked signals, it cannot impact the watermarked embedder.
>
> We thank the reviewer for pointing out this confusion. We clarify how the attribution loss impact the watermark embedder:
>
> In FakeMark, the attribution loss is computed on unwatermarked clean signals, allowing the decoder to learn intrinsic artifact patterns, same as a standard classifier-based attribution model.
>
> Meanwhile, FakeMark decoder is also trained with the watermark extraction loss on watermarked signals, which teaches it to identify the injected watermarks. These two losses jointly update the decoder. The watermark embedder is updated through the extraction loss, but the extraction loss depends on the decoder’s learned behavior, which is shaped by the attribution loss. **We revised line 221 to make this explicit**.
>
> **We also added embedding visualizations in Figure 4 in Appendix A.7** to further illustrate that clean and watermarked signals form coherent clusters and that the learned signatures align with system-specific artifact patterns.

---

> ### Author Response · Authors · 2025-11-28
> **Response to Reviewer z9Yp (Part 2)**
>
> **W2: lack of clarity**
>
> > Sect. 4.1 Line 252: evaluations on unseen architectures are beyond the scope of this work. It is even more than that: Evaluations on unseen models is simply impossible since the watermarking scheme needs the model for its training. This is a clear limitation of the proposal.
>
> **We added experiments in a new Sec. 4.3 to show that FakeMark can be extended to an open-set scenario without retraining**.
>
> Use case when new model rises: We discuss at of Sec. 4.3.1 that content owners can attribute previously unseen TTS systems by injecting an approximately matched signature, selected via nearest-neighbor search in the embedding space. This provides a lightweight approach to open-set attribution that does not require retraining FakeMark on data from the new model.
>
> Experiment setting: Unseen systems are assigned the watermark signature of the closest training system in the learned embedding space (illustrated in Appendix A.7), after which their signals are watermarked and processed by the FakeMark decoder following the same pipeline as prior experiments. Verification is performed by computing cosine similarity between decoder embeddings of reference and test signals and reporting equal error rate (EER) as the evaluation metric.
>
> Observation: As the results show, our proposed method remains effective for source verification on unseen architectures, and its behavior aligns with its robustness to the applied distortions. This indicates that FakeMark can leverage similarity among systems for marking the system in the open-set setting. This is further supported by Figure 3-5, where subsystems with related artifact patterns form close clusters.
>
> ---
>
> **W3 - lack of security**
>
> We agree with the reviewer that security is essential when applying the proposed scheme. The use of watermark removal attacks in our experiment (the bottom rows in Table 1 & 2) was intended to gauge the security. Furthermore, the “averaging attack” is based on the paper of Yang et al.
>
> We believe that the robustness and security of watermarking may be better approached in a more organized manner (as mentioned in [1]), wherein security may be, for example, enhanced by adding cryptographic primitives upon a robust watermarking scheme. Or, separate studies like the paper the reviewer mentioned should be done to investigate the vulnerability of machine-learning based watermarking schemes.
>
> In this work, our primary focus is on robustness to real-world audio distortions. **We have revised the manuscript to clarify this scope and to explicitly acknowledge security as a current limitation of FakeMark**.
>
> [1] I. Cox, G. Doërr, and T. Furon, “Watermarking is not cryptography,” In: Shi, Y.Q., Jeon, B. (eds) Digital Watermarking. IWDW 2006. Lecture Notes in Computer Science, vol 4283. Springer, Berlin, Heidelberg.
>
> ---
>
> **W4 - Unfair benchmark - Overwriting and Averaging**
>
> We thank the reviewer for pointing out this important issue. We agree that using the publicly available pre-trained AudioSeal and Timbre models as Overwriting or Averaging attacks is unfair, since these same models are also retrained as baselines in our evaluation.
>
> In the revised manuscript, for **overwriting attacks, we replaced them with other publicly available watermarking systems (WavMark and SilentCipher)**. For the **averaging attack, we used all possible evaluated models (baseline AudioSeal, both FakeMarks) to calculate the averaged watermarks**. All corresponding results (Tables 1, 2, and 4) and analysis have been updated accordingly.

---

> ### Author Response · Authors · 2025-11-28
> **Response to Reviewer z9Yp (Part 3)**
>
> **W4 - Unfair benchmark - AudioSeal**
>
> We confirm that our evaluation is correct.
>
> To address the discrepancy between our paper and the AudioSeal paper, we conducted two experimental analyses focusing on (1) speech quality and (2) robustness to speed perturbations.
>
> (1) We used our evaluation script to calculate speech quality of our retrained AudioSeal and pre-trained AudioSeal.
>
> Table R1 — Speech quality measurements on differently trained AudioSeals
> | Model                         | SI-SNR | PESQ | ViSQOL |   PQ  |
> |-------------------------------|--------|------|--------|-------|
> | Our retrained AudioSeal-4bits           | 36.49  | 4.55 | 4.98   | 6.78  |
> | Pre-trained AudioSeal-16bits  | 28.13  | 4.38 | 4.92   | 6.73  |
> | AudioSeal paper               | 26.00  | 4.47 | 4.83   |   -   |
>
>
> This shows:
>
> 1. The evaluated pre-trained model achieves SI-SNR values similar to those reported in the original AudioSeal paper, confirming that our evaluation script is correct.
>
> 2. Our retrained AudioSeal indeed achieves higher speech quality (including SI-SNR) than the pre-trained one.
>
> We hypothesize this is because our retrained model uses a smaller message length (4 bits vs. 16 bits). A smaller message imposes less payload on the generator and therefore increases perceptual quality.
>
> (2) We evaluated pre-trained AudioSeal’s bitwise accuracy and deepfake attribution accuracy using MLAAD_v5 test set.
>
> Table R2 — Attribution performance of pre-trained 16-bit AudioSeal
> | Distortion  | Bit-wise Accuracy | Attribution Accuracy |
> |--------------------|-------------------|-----------------------|
> | Pitch              | 0.54              | 0.14                  |
> | Speed              | 0.55              | 0.18                  |
> | Noise              | 0.86              | 0.79                  |
> | SpeechTokenizer    | 0.50              | 0.05                  |
> | FACodec            | 0.50              | 0.06                  |
> | WavTokenizer       | 0.50              | 0.09                  |
> | HiFi-GAN           | 0.50              | 0.06                  |
> | Vocos              | 0.52              | 0.10                  |
> | BigVGAN            | 0.77              | 0.67                  |
>
>
> While this evaluation is not directly comparable to the results in our paper (due to differences in training data and message length), it nevertheless shows that the pre-trained model is not robust to many perturbations.
>
> Also, in the original AudioSeal paper (Table 3), the reported robustness to a speed ×1.25 attack (0.99) corresponds to binary detection accuracy, not bitwise or attribution accuracy. As described in their Sec. 3.5, detection is an easier task than attribution, the latter requires recovering all injected bitstring messages correctly. AudioSeal attribution performance is reported in their paper Table 4, the reported average accuracy is 0.62.
>
> Speed-based removal attacks on AudioSeal have also been reported in other studies: for example, Table 1 in [1] and Table 1 in [2].
>
> [1] P. O’Reilly, Z. Jin, J. Su, and B. Pardo, “Deep Audio Watermarks are Shallow: Limitations of Post-Hoc Watermarking Techniques for Speech,” Apr. 15, 2025, arXiv: arXiv:2504.10782. doi: 10.48550/arXiv.2504.10782.
>
> [2] Y. Liu, L. Lu, J. Jin, L. Sun, and A. Fanelli, “XAttnMark: Learning Robust Audio Watermarking with Cross-Attention,” Feb. 07, 2025, arXiv: arXiv:2502.04230. doi: 10.48550/arXiv.2502.04230.

---

> ### Author Response · Authors · 2025-11-28
> **Response to Reviewer z9Yp (Part 4)**
>
> **W4**
> > The four techniques (FakeMark A and T, AudioSeal, Timbre) have strong differences in speech quality. I consider that a SISNR lower than 20dB is a dead-end. FakeMark^T is simply not useable. Moreover, comparing the robustness of watermarking scheme with much perceptibility differences is meaningless.
>
> We thank the reviewer for the comment. Reviewer FUV6 also questioned the quality degradation when using FakeMark$^T$. Even the Timbre itself obtained an SI-SNR around 20. We’ve tried a few ways to improve the SI-SNR and found that, while FakeMark$^T$’s perceptuality can be improved by emphasizing perceptual losses, this inevitably weakens its performance under distortions.
>
> Since FakeMark$^A$ and Audioseal maintain reasonably good SI-SNRs, we hypothesize that the degradation may rise from our Timbre retraining process. Due to time and page constraints, we were not able to explore more complex architectural changes or more extensive balancing of loss weights during training.
>
> ---
>
> **Q1: On random watermark message**
>
> Using random messages prevents the decoder from collapsing into relying solely on deepfake artifacts. We noticed that when we force the watermark message to always match the system label during training, the generator fails to learn meaningful watermarking embedding representations, and the FakeMark decoder degenerates into a purely artifact-based multi-class classifier model. In this case, adding or removing the watermark has almost no effect on attribution performance. Random messages are therefore essential for ensuring that the model actually learns artifact-correlated watermarks rather than depending exclusively on artifacts.
>
> ---
>
> **Q1: On attribution loss and watermark extraction loss**
>
> The attribution loss is used only to help the decoder learn the intrinsic artifact differences across TTS systems, it is not meant to maximize attribution accuracy on its own. This design encourages the injected watermark to become the primary attribution cue. As a result, FakeMark maintains robustness even when artifact patterns are weak or absent, because the decoder can still rely on the injected signature.
>
> ---
>
> **Q1: On alignment of watermarks and artifacts**
>
> **We added embedding space visualizations in Figure 3-6 in Appendix A.7** to illustrate the alignment between watermarks with artifact structure. These plots show that:
>
> * Clean and watermarked utterances from the same architecture form tight, overlapping clusters.
>
> * This behavior is consistent in both the validation set (Figure 3) and test set (Figure 4-5).

---

### Official Review · Reviewer_Bkfp · 2025-10-27

**Soundness:** 3
**Presentation:** 2
**Contribution:** 2
**Rating:** 4
**Confidence:** 4

**Summary:**

The paper introduces FakeMark, a novel watermarking framework for deepfake speech attribution. By injecting artifact-correlated watermarks, this design allows a detector to attribute the source system by leveraging both injected watermark and intrinsic deepfake artifacts, remaining effective even if one of these cues is elusive or removed. Experimental results show that it improves attribution robustness and generalization in challenging scenarios.

**Strengths:**

- The paper is generally well-organized and easy to read.
- The methodology is easy to follow.
- The paper presents the first systematic evaluation of deepfake speech attribution using both
watermarking-based and classifier-based models.

**Weaknesses:**

- The novelty of the proposed method appears to be limited. The method is largely an integration of existing techniques, namely watermarking method (e.g., Timbre) and the SSL-based classifier (MMS-300M). The framework's performance seems heavily dependent on the robustness of the chosen watermark and the accuracy of the classifier, rather than the proposed “watermarked artifacts” integration strategy itself. Therefore, the **added value** of the integration strategy itself is not immediately clear.
- The evaluation of the framework’s performance appears incomplete. The authors investigate scenarios where *either* the watermark is removed *or* the seen deepfake artifacts patterns are absent, but they do not address the more comprehensive scenario where both are removed simultaneously. The effectiveness of the method in this combined-attack setting remains unclear, which weakens the paper's overall robustness and claims.
- Some important details about the proposed approaches are not mentioned or explained in the paper.
   - Could the authors provide a more detailed explanation of the claimed "artifact-correlated watermark" design? Based on the paper's description, the watermark w still appears to be a class index w ∈ {1, . . . , C}, where C is the total number of deepfake systems. It is not immediately clear how this design is actually "artifact-correlated" as claimed.
  - The training process for the "Detector" module is not mentioned. How does its training differ from the "detector" in classifier-based attribution methods or the "decoder" in traditional watermarking methods? This distinction seems critical to understanding the novelty of the proposed approach.
  - Regarding the "Attribution loss," the authors state: "It is computed as the cross-entropy between the ground-truth deepfake system label and the detector’s predicted class probabilities over an unwatermarked clean signal." If this loss is indeed computed on an unwatermarked clean signal, it seems identical to training a standard classifier-based attribution model. Can the authors please confirm if this is correct, or if this loss should be computed on a watermarked signal instead?
  - In the experimental setup, the terms "seen artifacts," "unseen artifacts," "seen architectures," and "unseen architectures" are used without explicit definitions. This terminology is ambiguous and confusing.
- Some typos:
  - In line 88: “watermarking-” should be “watermarking-based”

**Questions:**

- What is the framework's effectiveness in the more challenging scenario where both the watermark and the seen deepfake artifacts are removed simultaneously?
- The paper evaluates robustness against various distortions in isolation (e.g., only compression, or only noise). However, a more realistic scenario would involve composite or sequential attacks (e.g., codec compression followed by noise or pitch-shift). How does the proposed watermark perform under such combined distortions? Were any experiments conducted for this?

---

> ### Author Response · Authors · 2025-11-28
> **Response to Reviewer Bkfp (Part 1)**
>
> We thank the reviewer for their careful evaluation and valuable comments. Below, we address each of the points raised.
>
> ---
>
> **W1: Novelty of the integration strategy**
>
> We agree with the reviewer that the individual components of FakeMark are based on existing techniques. However, the novelty of FakeMark, as noted by Reviewers FUV6, zpYx and z9Yp, lies in how these components are formulated and jointly trained to create a hybrid attribution system in which watermarking and artifact-based classification reinforce each other within a unified framework.
>
> Our experiments demonstrate that the performance gains cannot be attributed solely to the robustness of the underlying watermark generator (e.g., FakeMark$^A$ outperforming AudioSeal under neural-codec distortions) nor to the accuracy of the classifier backbone (e.g., FakeMarks outperforming MMS-300M in cross-dataset evaluations). These improvements arise specifically from FakeMark’s integration of system signatures (i.e., watermarks) with artifact-based attribution, rather than from the strengths of the individual components alone.
>
> Hence, we sincerely hope that the reviewer can re-evaluate the novelty of this work.
>
> ---
>
> **W2 and Q1: Evaluation on both watermark and artifacts are removed or absent**
>
> We thank the reviewer for highlighting this important scenario, but it is kindly noted that Table 2 and Sec.4.2.2 cover the evaluation case suggested by the reviewer.
>
> Specifically, in the Removal Attack panel of Table 2, the experimental systems are tested on unseen deepfakes (hence likely unseen artifacts) and watermark-removal attacks. The results show that both FakeMark variants sustain reliable attribution under averaging-based removal and remain more robust than classifier baselines under overwriting, indicating that the injected system signatures (i.e., watermarks) continue to support reliable attribution even on unseen deepfakes and malicious removal attacks.
>
> ---
>
> **W3: Details of description:**
>
> > Could the authors provide a more detailed explanation of the claimed "artifact-correlated watermark" design? Based on the paper's description, the watermark w still appears to be a class index w ∈ {1, . . . , C}, where C is the total number of deepfake systems. It is not immediately clear how this design is actually "artifact-correlated" as claimed.
>
> Sure! Although the watermark is indexed by a class label, we do not use this numeric index as the watermark itself. Instead, similar to other recent ML-based watermarking methods like AudioSeal [1] and XAttnMark [2], we use this index only to retrieve a learned embedding from a network module (i.e, the message processor) that is jointly updated with the whole system. This watermark embedding is updated by both attribution loss and watermark extraction loss to be artifact-correlated, and later it is injected into the latent space to reconstruct the watermarked signal.
>
> **We have updated related lines in Sec.3 to clarify this mechanism**. In addition, **we provide embedding space visualizations in Figures 4–6 in Appendix A.7** to illustrate how clean and watermarked signals cluster according to these learned, artifact-correlated embeddings.
>
>
> [1] R. San Roman, P. Fernandez, H. Elsahar, A. Defossez, T. Furon, and T. Tran, “Proactive detection of voice cloning with localized watermarking,” in Proc. ICML, 2024.
>
> [2] Y. Liu, L. Lu, J. Jin, L. Sun, and A. Fanelli, “XAttnMark: Learning robust audio watermarking with cross-attention,” in Proc. ICML, 2025.

---

> ### Author Response · Authors · 2025-11-28
> **Response to Reviewer Bkfp (Part 2)**
>
> **W3: Details of description:**
>
> > The training process for the "Detector" module is not mentioned. How does its training differ from the "detector" in classifier-based attribution methods or the "decoder" in traditional watermarking methods? This distinction seems critical to understanding the novelty of the proposed approach.
>
> The relevant components were described in Sec. 3.2, but we agree that the terminology may cause confusion. To clarify, **we have standardized the terms in the revised manuscript** as follows:
>
> * Detector – binary deepfake detector (real vs. fake)
> * Classifier – multi-class attribution model predicting the generating system
> * Decoder – watermark decoder that extracts the injected watermarks
> * Injection / Extraction – adding and recovering watermark signals
>
> Our proposed system uses a “Decoder”, but it blends the idea of decoder in conventional watermarking and the classifier in attribution.
>
> In traditional watermarking, decoders are trained to recover a bitstring from any input waveform. They do not use or learn any information about artifact patterns of the source model. **We highlight this distinction in the revised text near line 215**.
>
> In classifier-based attribution, models are trained and evaluated on clean, unwatermarked signals and perform standard multi-class classification to predict which system generated the signal.
>
> FakeMark’s decoder is related to but differs from both. FakeMark’s decoder predicts a probability distribution over system classes, similar to a classifier, but it is trained under a watermark extraction loss that encourages it to identify which watermark was injected to reconstruct the speech signal.
>
> This terminology is now reflected consistently in the revised manuscript.
>
> ---
>
> > Regarding the "Attribution loss," the authors state: "It is computed as the cross-entropy between the ground-truth deepfake system label and the detector’s predicted class probabilities over an unwatermarked clean signal." If this loss is indeed computed on an unwatermarked clean signal, it seems identical to training a standard classifier-based attribution model. Can the authors please confirm if this is correct, or if this loss should be computed on a watermarked signal instead?
>
> Yes, this is correct. In FakeMark, the attribution loss is computed on unwatermarked clean signals, allowing the decoder to learn artifact patterns of each TTS system, this is same as a standard classifier-based attribution model. As shown in Table 4 (“No watermark”), the decoder alone functions as a standalone classifier similar to MMS-300M in Tables 1 & 2.
>
> Meanwhile, FakeMark decoder is also trained with the watermark extraction loss on watermarked signals, which teaches it to identify the injected watermarks. These two losses jointly update the decoder, so its overall behavior is not identical to a classifier: FakeMark still predicts multiclass probabilities, but each class now leverages both intrinsic artifact cues and the injected watermarks. **We added embedding visualizations in Figure 3** to further illustrate that clean and watermarked signals form coherent clusters and that the learned signatures align with system-specific artifact patterns.
>
> ---
>
> > In the experimental setup, the terms "seen artifacts," "unseen artifacts," "seen architectures," and "unseen architectures" are used without explicit definitions. This terminology is ambiguous and confusing.
>
> We thank the reviewer for pointing this out. In the revised manuscript, **we have replaced these ambiguous terms in the experimental setup and subsection titles**. We now explicitly define “unseen” at its first occurrence in Sec.4.2.1 to mean cases where the artifact patterns of a system are not present in the training set.
>
> ---
>
> > In line 88: “watermarking-” should be “watermarking-based”
>
> Thanks! **We have corrected this typo, and we also fixed a similar occurrence in the Conclusion section**.
>
> ---
>
> **Q2: Results on challenging sequential distortions**
>
> We conducted related results, due to page limit, **we added them to Appendix A.8 in the revised manuscript**.
>
> Setup: We first applied a neural codec or vocoder, followed by a second distortion such as signal transformations or adversarial attacks.
>
> Observation: We observed similar trends as in Table 1&2. All systems degrade most when WavTokenizer is used, and the combination of WavTokenizer with MUSAN noise yields the lowest accuracy. While classifier baselines remain strong in-domain dataset but fail cross-domain dataset, both Timbre and FakeMark$^T$ maintain stable performance in both test sets.

---

### Official Review · Reviewer_FUV6 · 2025-10-30

**Soundness:** 3
**Presentation:** 2
**Contribution:** 2
**Rating:** 4
**Confidence:** 3

**Summary:**

The paper proposes FakeMark, a hybrid framework for deepfake speech attribution. It aims to address the key weaknesses of existing methods: the poor generalization of classifiers and the vulnerability of watermarks to distortions. The core contribution is the concept of an 'artifact-correlated watermark,' where the injected watermark is associated with the intrinsic artifacts of a specific deepfake system. A detector is then trained to recognize both the watermark and the artifacts simultaneously. The authors claim this dual approach improves robustness against distortions (by falling back on artifacts if the watermark is removed) and generalization to new datasets (by relying on the watermark).

**Strengths:**

- The paper's originality lies in its novel problem formulation of a hybrid attribution system. The core idea of injecting an "artifact-correlated watermark" —combining watermark detection and artifact classification into a single, dual-signal framework—is a creative combination of existing concepts.
- The authors perform a comprehensive evaluation. The method is tested under a wide array of challenging distortions, including neural codecs, vocoders, and specific watermark removal attacks.

**Weaknesses:**

The results in Table 3 suggest a strong trade-off: the most robust models (FakeMark-S, Timbre) have the worst audio quality (PESQ 2.83 / 2.97). This implies the robustness is achieved by injecting a stronger, more perceptible watermark. Is this a fundamental limitation of the spectrogram-based approaches?

**Questions:**

Please provide solutions to the degraded perpectuality.

---

> ### Author Response · Authors · 2025-11-28
> **Response to Reviewer FUV6**
>
> We thank the reviewer for their careful evaluation and valuable comments. Below, we address each of the points raised.
>
> ---
>
> **W1: Limitation on trade-off between audio quality and robustness.**
>
> We agree that there is an inherent trade-off between watermark perceptual quality and robustness, but it applies broadly to watermarking systems, not limited to FakeMark or other spectrogram-based approaches.
>
> As discussed in Chapter 18.3 of [1], regarding fidelity under compression: (1) if a watermark is truly imperceptible, it will likely be removed or degraded by lossy compression and (2) watermarks that survive compression will cause increasingly perceptible differences. This argument is reflected in our experiment results.
>
> [1] M. L. Miller et al., “A Review of Watermarking Principles and Practices,” in Digital Signal Processing in Multimedia Systems, ch. 18, pp. 461–485, 1999.
>
> ---
>
> **Q1: Solutions to the degraded speech quality.**
>
> We conducted additional experiments to improve the speech quality of FakeMark$^T$. Specifically, we:
> 1. Increase the weights for one of the Perceptual losses (the HiFi-GAN loss: spectrogram reconstruction loss + GAN loss) from 1.0 (used in original manuscript) to 7.0 (used in revised manuscript) and 15.0.
> 2. Only enable distortions after 100k steps during training, which we found to reduce watermark energy level, resulting in higher speech quality.
>
> Results are shown below.
>
> Table R1 — Attribution accuracy on selected distortions
> | Model                | clean | facodec | speed | pitch |
> |----------------------|-------|---------|-------|-------|
> | FakeMark$^T$ (λₚ = 1)   | 1.00  | 0.99    | 1.00  | 1.00  |
> | FakeMark$^T$ (λₚ = 7)   | 1.00  | 0.85    | 1.00  | 0.96  |
> | FakeMark$^T$ (λₚ = 15)  | 1.00  | 0.30    | 0.70  | 0.70  |
>
> Table R2 — Speech quality
> | Model     | SI-SNR ↑ | PESQ ↑ | ViSQOL ↑ | PQ ↑ |
> |-----------|----------|---------|----------|-------|
> | λₚ = 1    | 14.97    | 2.83    | 4.41     | 6.18  |
> | λₚ = 7    | 19.14    | 3.09    | 4.64     | 6.50  |
> | λₚ = 15   | 22.89    | 3.12    | 4.51     | 6.37  |
>
> We found that, while FakeMark$^T$’s perceptuality can be improved by emphasizing perceptual losses, this inevitably weakens its performance under distortions. **We have updated all results related to FakeMark$^T$ with the new results (λₚ = 7)**.
>
> Due to time and page constraints, we were not able to explore more complex architectural changes or more extensive balancing of loss weights during training. However, following [1], another direction is to selectively control the speech region where the watermark is injected. In our setting, this could involve constraining which frequency regions contribute to the frequency-magnitude loss so that watermark energy is concentrated in less/more perceptually sensitive frequency bands. We have added this point to the Conclusion as future work.

---

### Official Review · Reviewer_zpYx · 2025-10-30

**Soundness:** 2
**Presentation:** 3
**Contribution:** 3
**Rating:** 4
**Confidence:** 3

**Summary:**

The paper tackles the task of model attribution of audio deepfakes: given a synthesized audio, predict which model it was generated with. The typical approach is to train a network that maps an audio deepfake to its corresponding model category. Instead, the proposed method, FakeMark, takes inspiration from watermarking and adds auxiliary information to the signal to ease detection. However, unlike traditional watermarking, the injected watermark depends on the model. This approach can be understood as amplifying the generators' fingerprints through learned watermark embeddings. The authors show that this approach gives better attribution results than independent fingerprints, although at a slight cost in intelligibility.

My rating is close to borderline. Justification: The paper proposes an interesting point in the design space of model attribution systems, but with an unclear practical use case.

**Strengths:**

- The approach is interesting and provides a novel point in the design space of model attribution systems. Framing it as "fingerprint amplification" rather than traditional watermarking would better capture its actual mechanism.
- The paper is well executed with experiments on well-established benchmarks and methods. The cross-dataset generalization results demonstrate that watermarks help when artifacts shift.
- The paper provides a comprehensive evaluation of robustness to various distortions (codecs, vocoders, signal processing, removal attacks).

**Weaknesses:**

- It's unclear what the envisioned use case is for such an approach. One goal of model attribution is to identify the vendors who produce a certain deepfake, in order to hold them accountable. However, applying the proposed method across vendors requires a centralized system with which all vendors agree with (since the watermarking model has to be shared). This seems like a very strong assumption. Alternatively, for attribution of in-the-wild deepfakes, where such cooperation cannot be expected, the detector reduces to an artifact-based classifier, making the watermarking component superfluous.
- The proposed method misses some of the key characteristics of watermarking (e.g., embedding an independent and verifiable information into the signal). For this reason, calling it "a watermarking framework" may be misleading; instead, it may be more accurately regarded as a fingerprint-amplifying method.

**Questions:**

- See weaknesses.
- How does the similarity of the model waterkmarking embeddings looks like? Do we get similar watermarking embeddings for similar models? This would validate the "artifact-correlated" claim.
- How does the approach scale with the number of models to be attributed? Would the embedding space get crowded and the artifacts start overlapping between models?
- The proposed approach assumes the set of models fixed (closed set). If a new model arises, one has to retrain FakeMark. This seems impractical. Could the approach be extended to an open-set setting, for instance by first training a model encoder (e.g., via metric learning) and then using its embeddings as watermarking embeddings?

---

> ### Author Response · Authors · 2025-11-28
> **Response to Reviewer zpYx**
>
> We thank the reviewer for their careful evaluation and valuable comments. Below, we address each of the points raised.
>
> ---
>
> **W1&Q1: Use case for the proposed approach**
>
> In the revised manuscript, **we have clarified the intended use case** (line 27-31, line 88-94). Our goal is not to propose a cross-vendor, globally coordinated watermarking standard. Instead, FakeMark targets proactive, content-owner–controlled attribution, where the party generating or distributing synthetic speech has the ability to inject system-specific signatures as watermarks.
>
> ---
>
> **W2&Q1: Framing of the proposed approach**
>
> In the revised manuscript, **we have reframed the terminology and presentation of FakeMark** to better reflect its actual mechanism.
>
> We investigated a few papers proposing similar but subtly different taxonomies of the watermark applications, and we decided to adopt the term “system signature” as the primary term when describing the injected watermarks. This terminology aligns with the definitions in [1] where signatures refer to watermarks used to identify the owner or producer of the content, while fingerprints are watermarks used to identify individual buyers or consumers. Since FakeMark is designed to attribute the generative model rather than individual users, we choose “signature”.
>
> Another reason that we didn’t use fingerprinting is that, fingerprinting seems to also refer to the application of summarizing and retrieving each individual audio sample [2], similar to the thumb fingerprint.
>
> **We have updated the manuscript accordingly, including revisions to the title, abstract, introduction, and relevant sections** throughout the paper to ensure consistent framing.
>
> [1] M. L. Miller et al., “A Review of Watermarking Principles and Practices,” in Digital Signal Processing in Multimedia Systems, ch. 18, pp. 461–485, 1999.
>
> [2] J. Haitsma and T. Kalker, “A highly robust audio fingerprinting system.,” in Proc. ISMIR, 2002, pp. 107–115.
>
> ---
>
>
> **Q2 & Q3: Visualizations of embedding space**
>
> **We added visualizations of the learned embedding space in Figures 3-6 in Appendix A.7** to address these questions.
>
> On embedding similarity (Q2): We notice that clean and watermarked utterances from the same architecture cluster tightly and occupy similar regions in the embedding space (Figure 4).
>
> Based upon Figure 4,  we draw Figure 5 wherein we use additional colors to distinguish systems that share the same architecture but differ in configuration (e.g., systems using the VITS-based TTS architecture but trained on different speakers’ data). We observe that these variants remain within their architecture-level cluster.
>
> On scaling with more attribution models (Q3): We retrained FakeMark$^A$ with all 24 original system labels as separate classes and visualized the resulting embedding space in Figure 6. Most systems continue to form clear and distinguishable clusters, but as the number of classes increases, clusters begin to exhibit partial overlap. This is not surprising because some of the “24” systems adopt the same TTS architecture (e.g., VITS variants & Tracotron 2 variants).
>
> ---
>
> **Q3 & Q4: Scale to new models in open-set setting**
>
> **We added both analysis and experiments in the revised Sec. 4.3** to demonstrate how FakeMark can be extended to an open-set scenario without retraining.
>
> The idea is that, when a new TTS model is to be processed by FakeMark, we inject an approximately matched signature as watermark. This signature is selected via nearest-neighbor search in the embedding space (Figure 3). This provides a lightweight approach to open-set attribution that does not require retraining FakeMark on data from the new model. This idea is based on what the reviewer suggested – we leverage the embeddings learned by FakeMark.
>
> The experiment results were added to Table 5 in Sec.4.3.2. We perform embedding similarities comparisons on reference and test signals, where both signals are watermarked with the signature from the training system whose embedding centroid is closest to these signals’ unseen system in the learned space.
>
> As the results show, our proposed method remains effective for source verification on unseen architectures, and its behavior aligns with its robustness to the applied distortions. This indicates that FakeMark can leverage similarity among systems for marking the system in the open-set setting. This is further supported by Figure 3-5, where subsystems with related artifact patterns form close clusters.

---

### Meta-Review · Area_Chair_FW2h · 2025-12-08

**Summary:**

The paper proposes FakeMark, a hybrid framework for deepfake speech attribution that combines watermarking with artifact-based classification. The core idea is to inject system signatures that correlate with the intrinsic artifacts of the generative model.

While reviewers acknowledged the potential of the hybrid approach, there was a consensus that the submission suffered from significant methodological and practical flaws. The primary concerns included unfair experimental comparisons against baselines, fundamental security vulnerabilities, and a severe trade-off between audio quality and robustness. Despite the authors' active engagement during the rebuttal, the core limitations regarding security and the practicality of the proposed use case remain unresolved. Therefore, I recommend this paper for rejection.

**Reviewer Concerns:**

During the rebuttal, the authors successfully addressed several specific concerns. They resolved terminology confusion raised by Reviewers zpYx and Bkfp by redefining deepfake detectors as attribution classifiers and adding open-set experiments.

However, critical concerns remain outstanding, most notably the security vulnerability highlighted by Reviewer z9Yp, where the use of a constant watermark in the latent space makes the system susceptible to simple averaging attacks.

**Reviewer Scores:**

The reviewer scores may largely remain unchanged despite the active discussion. Reviewers zpYx, FUV6, and Bkfp would likely maintain their scores because, although the authors clarified definitions and added experiments, the fundamental issues regarding the practicality of the use case, the limited novelty of the integration, and the confirmed degradation of robustness when ensuring high audio quality were not fully resolved. Reviewer z9Yp might slightly increase their score to acknowledge the correction of the unfair benchmarks. But given the strong technical critique regarding the inherent insecurity of the fixed latent watermark, Reviewer z9Yp would likely still recommend rejection.

---

### Decision · Program_Chairs · 2026-01-26

Reject